# Reactivated endogenous retroviruses promote protein aggregate spreading

Shu Liu[1,6,7], Stefanie-Elisabeth Heumüller[1,7], André Hossinger[1], Stephan A. Müller [2], Oleksandra Buravlova[1], Stefan F. Lichtenthaler [2,3,4], Philip Denner[1] & Ina M. Vorberg [1,5] ✉

Prion-like spreading of protein misfolding is a characteristic of neurodegenerative diseases, but the exact mechanisms of intercellular protein aggregate dissemination remain unresolved. Evidence accumulates that endogenous retroviruses, remnants of viral germline infections that are normally epigenetically silenced, become upregulated in neurodegenerative diseases such as amyotrophic lateral sclerosis and tauopathies. Here we uncover that activation of endogenous retroviruses affects prion-like spreading of proteopathic seeds. We show that upregulation of endogenous retroviruses drastically increases the dissemination of protein aggregates between cells in culture, a process that can be inhibited by targeting the viral envelope protein or viral protein processing. Human endogenous retrovirus envelopes of four different clades also elevate intercellular spreading of proteopathic seeds, including pathological Tau. Our data support a role of endogenous retroviruses in protein misfolding diseases and suggest that antiviral drugs could represent promising candidates for inhibiting protein aggregate spreading.

Neurodegenerative diseases are associated with the aberrant folding of host-encoded proteins into insoluble, highly structured beta sheet-rich protein complexes, termed amyloid. Misfolding of proteins such as the microtubule-binding protein Tau is associated with highly prevalent Alzheimer's disease (AD) and other tauopathies. In AD, Tau deposition precedes grey matter atrophy, arguing that misfolded Tau is a major driver of pathogenesis[1]. Several different proteins such as TDP-43 or FUS accumulate in the central nervous system of patients suffering from amyotrophic lateral sclerosis (ALS) or frontotemporal lobar degeneration (FTLD)[2]. While mutations in aggregation-prone proteins account for some cases of familial neurodegenerative diseases, the etiologies of spontaneous diseases are unknown. Protein misfolding occurs through a process of templated conversion, in which small oligomers of misfolded proteins eventually fold into amyloid fibrils capable of templating their aberrant fold onto soluble proteins of the same kind. Protein aggregation appears to proceed along neuroanatomical projections, arguing that intercellular dissemination and propagation of protein misfolding underly disease progression[3]. This process resembles the spreading of prions, infectious protein aggregates composed of PrP that are the causative agents of transmissible spongiform encephalopathies (TSEs). Indeed, a growing number of studies using cellular models or animals provide substantial evidence for the spreading of protein aggregates between cells and within tissues[4,5]. Small seeds of aggregated proteins can be either directly released by affected cells or transmitted to bystander cells via direct cell contact[6]. Proteopathic seeds capable of inducing protein aggregation in recipient cells can also be packaged into

[1]German Center for Neurodegenerative Diseases Bonn (DZNE), Venusberg Campus 1/ 99, 53127 Bonn, Germany. [2]German Center for Neurodegenerative Diseases (DZNE), Munich, Germany. [3]Neuroproteomics, School of Medicine, Klinikum rechts der Isar, Technical University of Munich, 81675 Munich, Germany. [4]Munich Cluster for Systems Neurology (SyNergy), Munich, Germany. [5]Department of Neurology, Rheinische Friedrich-Wilhelms-Universität Bonn, Germany. Venusberg-Campus 1, 53127 Bonn, Germany. [6]Present address: German Federal Institute for Risk Assessment (BfR), German Centre for the Protection of Laboratory Animals (Bf3R), Max-Dohrn-Straße 8-10, 10589 Berlin, Germany. [7]These authors contributed equally: Shu Liu, Stefanie-Elisabeth Heumüller. ✉e-mail: ina.vorberg@dzne.de

extracellular vesicles (EVs), which are normally secreted by cells for intercellular communication.

We have recently shown that viral glycoproteins such as the vesicular stomatitis virus protein G or SARS CoV-2 spike S expressed by protein aggregate-bearing cells can mediate efficient intercellular contact with bystander cells, resulting in protein aggregate induction[7]. Moreover, viral glycoprotein decoration of EVs from donor cells harboring proteopathic seeds composed of Tau, the yeast Sup35 prion domain NM or prion protein PrP strongly increased the EVs' aggregate-inducing capacity in recipient cells. Thus, viral glycoproteins expressed during infection could act as "address codes" that enable delivery, receptor binding, efficient uptake, and cytosolic release of proteopathic cargo into recipient cells.

Viral genes are not only encoded by exogenous viruses invading mammalian cells but are also remnants of mammalian germline infections that happened millions of years ago. Approximately 8–10% of human and mouse genomes consist of retroviral elements. Endogenous retroviruses (ERVs) share a common genome architecture with their exogenous counterparts, in which the coding regions for the capsid proteins (*gag*), reverse transcriptase, integrase and protease (*pol*) and envelope glycoprotein (*env*) are flanked by long terminal repeats. The majority of ERVs are reduced to proviral fragments, and only a few are intact or at least contain full open-reading frames that are transcribed and/or translated[8]. ERVs are subject to tight control by epigenetic modifications that repress transcription. Failure to silence ERVs is associated with cancer as well as autoimmune, inflammatory, and neurodegenerative disorders[9,10]. Importantly, several human ERV (HERV) members are upregulated in the brains of tauopathy and ALS patients[11–17]. No HERV-derived infectious virions have so far been detected, but HERV-expressing human cell lines can produce viral-like particles[18].

Murine ERVs of the Moloney leukemia virus (MLV) clade have integrated into the germline of ancestors millions of years ago. Some inbred mouse lines constitutively generate infectious MLV particles[19]. In most inbred mouse lines, however, the propensity of individual MLV loci to produce infectious virions is low. Still, restoration of ERVs by recombination events between the dozens of distinct loci of MLV subgroups[20] in immunocompromised mice induces ERV viremia[21].

Here we uncover that activation of endogenous MLVs strongly affects the spreading of proteopathic seeds between cells. By studying the spreading behavior of cytosolic protein aggregates composed of a yeast prion domain in cell culture, we demonstrate that the reactivation of MLVs strongly increases intercellular aggregate transmission. Reconstitution of HEK donor cells propagating cytosolic NM yeast prions or Tau aggregates with MLV Env was sufficient to promote protein aggregate transfer between cells but was increased when additional viral gene products were present. Targeting receptor binding or viral protein maturation drastically reduced intercellular proteopathic seed spreading. Further, the expression of diverse HERV Env glycoproteins also increased protein aggregate dissemination in cocultures. These findings raise the possibility that the derepression of ERVs accelerates the prion-like spreading of protein aggregates and suggests that ERVs represent potential therapeutic targets for disease intervention.

## Results

### Correlation of ERV expression and increased aggregate induction

For this study, we made use of our cell model which is based on the Sup35 domain NM. NM contains the prion domain of the *Saccharomyces cerevisiae* translation termination factor Sup35 that can form self-templating protein aggregates. The prion domain shares compositional similarity with prion-like domains of RNA-binding proteins FUS and TDP-43, known to form protein aggregates in ALS and FTLD[22]. Soluble NM expressed in mammalian cells

can be induced to aggregate by recombinant NM amyloid fibrils, resulting in cell populations that faithfully replicate NM aggregates over multiple passages[23]. Once induced, NM aggregates also transmit to bystander cells by direct cell contact or via EVs, thereby inducing ongoing NM aggregation[24,25]. Using our mouse neuroblastoma N2a Sup35 NM model system, we isolated N2a subclone s2E, with HA epitope-tagged Sup35 NM prion aggregates (NM-HA$^{agg}$) (Supplementary Fig. 1a). Interestingly, this clone outcompeted other clones in its aggregate-inducing capacity in recipient cells[25]. For simplicity, we here call this donor clone N2a NM-HA$^{agg}$. Surprisingly, its aggregate-inducing activity in coculture experiments strongly increased when cells were cultured over prolonged periods of time (Fig. 1a–c; Supplementary Fig. 1b). The increase in aggregate-inducing capacity was reproducible, occurring approximately between 7 and 16 passages post cryopreservation (approx. 32–72 days). Donor cells at high passage number retained their NM aggregate-inducing activity even when cryopreserved at passage 21 and subsequently taken into culture (Supplementary Fig. 1c). For convenience, donor cells cryopreserved at P1 or P21 were subsequently used for experiments for up to 6 passages if not otherwise noted. For simplicity, we refer to cell populations as early and late passage donors (EP and LP, respectively). Increased donor passage number also increased aggregate induction in recipients cultured with conditioned medium from donors. This effect was abolished when the conditioned medium was sonicated prior to addition to recipients, suggesting that aggregates might have been contained in EVs[25] that were destroyed by sonication (Supplementary Fig. 1d, e).

We previously demonstrated that NM aggregates are transmitted to bystander cells by EVs[25]. To test if this was also the case at the late passage, EVs from donors of different passages were purified by differential centrifugation. Strong aggregate induction was observed with EVs from late passage donors, demonstrating that EVs were involved in cell-free aggregate spreading (Fig. 1d, e). Sonication basically abolished the aggregate-inducing activity of EVs (Supplementary Fig. 1f, g). Increased induction was not due to increased EV secretion, as particle numbers did not change significantly over prolonged culture (Fig. 1f). EVs isolated from late passage donor cells also increased NM aggregate induction in primary cortical neurons, arguing that intercellular aggregate induction did not require mitotically active cells (Fig. 1g). No aggregate-inducing activity was associated with the supernatant fraction following ultracentrifugation, arguing that soluble factors such as cytokines unlikely played a major role in intercellular aggregate induction (Supplementary Fig. 2a). In line with this, real-time PCR using donor mRNAs did not reveal major differences in cytokine expression upon prolonged culture (Supplementary Fig. 2b).

To identify changes in the proteomes of donor cells that might contribute to protein aggregate spreading, we performed mass spectrometry analyses of total cell lysates (Fig. 1h) and donor EV fractions (Fig. 1i) at early and late passages post cryopreservation. Among the proteins increased in donor cells and EVs upon prolonged culture, we identified mouse endogenous MLV proteins to be highly increased (Supplementary Data 1). The partially overlapping open-reading frames of MLV code for polyproteins Gag (comprising nucleocapsid, capsid, p12, and matrix protein), Pol (reverse transcriptase, integrase, and protease) and for the envelope glycoprotein Env. They are transcribed as full genomic viral RNAs as well as subgenomic mRNAs coding only for *env* (Supplementary Fig. 3a–d). Prolonged cell culture increased mRNA levels coding for MLV *env* and *gag/pol* in both cell lysates and EVs (Fig. 1j). By contrast, transcripts for retroelements IAP, ETnI/II, LINE-1, and MusD remained largely unaffected (Supplementary Fig. 3e). Western blot analyses confirmed increased expression of Env and Gag in cell lysates and EV fractions from donors upon prolonged culture (Fig. 1k). The presence of capsid (CA) demonstrates Gag processing by the viral protease (Fig. 1k).

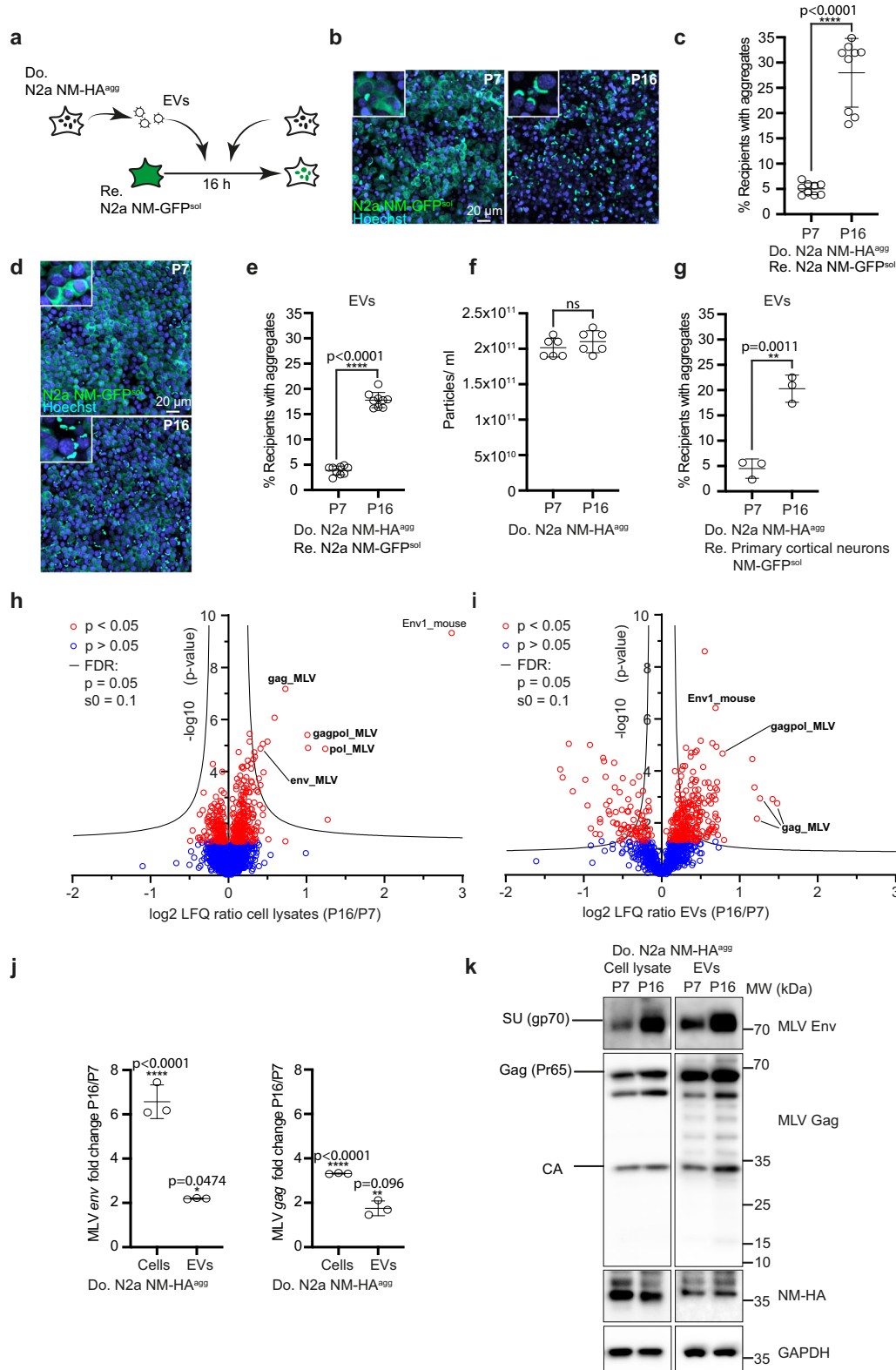

## Donor cells produce both active viral particles and EVs

The finding that MLV mRNA and proteins strongly increased in donor cells upon prolonged culture suggested that donor cells secrete active retroviruses, as has been observed for a few cell lines before[26,27]. In line with this, increased reverse transcriptase activity was observed upon the prolonged culture of donors (Fig. 2a). Dozens of MLVs proviruses populate the murine genome[20] and are

classified by their respective Env proteins as ecotropic, polytropic, and xenotropic, dependent on their receptor preferences. To test if the expression of ERVs had been triggered by NM aggregation, we performed quantitative real-time PCR on mRNA extracted from different N2a NM populations before and after exposure to recombinant NM fibrils. Aggregate induction had no influence on the induction of endogenous MLV subgroups, arguing that the

**Fig. 1 | Upregulation of murine endogenous retrovirus in donor cells increases intercellular aggregate induction. a** Experimental workflow. Donor N2a NM-HA$^{agg}$ clone passage 7 (P7) and 16 (P16) or donor EVs were added to recipient N2a NM-GFP$^{sol}$ cells. Analysis was performed 16 h post-exposure. **b** Confocal images of recipient and donor cells P7 or P16. Shown are Z-stacks. Nuclei were stained with Hoechst. Note that donors have not been stained for NM-HA. Insets: Close-ups. **c** Percentage of recipient cells with induced NM-GFP aggregates upon coculture with donors P7 and P16. **d** Confocal images of recipient cells exposed to EVs from donors P7 and P16. **e** Percentage of recipient cells with NM-GFP aggregates. **f** Particles isolated from the conditioned medium of donor cells P7 and P16 post-thawing were analyzed by nanoparticle tracking. **g** Primary neurons expressing soluble NM-GFP were exposed to donor EVs of early or late passage. Quantitative analysis of neurons with NM-GFP aggregates. **h** Volcano plot of total cell proteome of donor N2a NM-HA$^{agg}$ cells at lower and higher passage numbers. Proteins were

ranked according to their *P*-value and their relative abundance ratio (log2 fold change) in cells of P16 compared to cells of P7. **i** Volcano plot of proteomes of EVs derived from donor N2a NM-HA$^{agg}$ cells P16 versus P7. Proteins were ranked as above. **j** qRT-PCR analysis of *env* and *gag* mRNA of N2a NM-HA$^{agg}$ cells. Shown is the fold change in expression in donor cells P16 versus P7. **k** Increased MLV Env and Gag expression upon continuous cell culture of cells. Env and GAPDH were detected on the same blot, Gag, and NM-HA on a second blot. Shown are Env surface unit SU (gp70) and Gag polyprotein Pr65 and capsid (CA). All data are shown as the means ± SD from three (**g**, **j**), six (**f**), or nine (**c**, **e**) replicate cell cultures. Three (**c**, **e**–**j**) independent experiments were carried out with similar results. *P*-values calculated by two-tailed unpaired Student's *t*-test (**c**, **e**–**i**) or one-way ANOVA with Bonferroni´s multiple comparisons (**j**). ns: non-significant. Source data are provided as a Source Data file.

upregulation of MLV is likely influenced by other means (Supplementary Fig. 4a–f).

To demonstrate that donor cells produced infectious viruses, vesicle fractions derived from donors were added to murine Melan-a cells, a cell line permissive for MLV. Detection of Gag and Env by Western blot demonstrated that cells produced an infectious virus (Fig. 2b)[28]. We further tested if Vectofusin-1, a compound that increases viral interaction with cellular membranes, could enhance aggregate induction[29]. Aggregate induction was also increased when a conditioned medium was added to recipients in the presence of Vectofusin-1 (Supplementary Fig. 4g, h).

Interestingly, MLV virus-like particles can be formed that lack retroviral RNA[30]. To test which vesicle fractions released by the donor cells were NM-seeding competent, we separated EVs from viral particles by an Iodixanol velocity gradient previously used to separate HIV-1 virions from microvesicles (Fig. 2c)[31]. Fractions with the highest particle concentrations (fractions 2–6) harbored Alix and NM-HA, arguing that they contained EVs (Fig. 2d, e). Gag and Env were distributed throughout the gradient, with highest levels present in Alix-positive fractions (Fig. 2e). Reverse transcriptase (RT) activity was almost exclusively present in fractions 8–11, arguing that these fractions contained active virus (Fig. 2f). This was confirmed by electron microscopy, which revealed membranous 80–100 nm spherical particles with an electron-dense core, characteristic of γ-retroviral particles in fractions 9 and 10 (Fig. 2g). By contrast, vesicles in fractions 2 and 3 exhibited a cup-shaped morphology, characteristic of EVs in TEM. Next, we tested the NM aggregate-inducing activity of fractions by adding them to recipient N2a NM-GFP$^{sol}$ cells (Fig. 2h). Highest aggregate induction was associated with RT-negative EV fractions 2–6 (Fig. 2h). Interestingly, aggregate induction could be inhibited by antibodies against MLV Env in both coculture and EV experiments (Fig. 2i, j). We conclude that NM aggregate-seeding activity is mainly associated with the RT-negative EV fraction, and aggregate induction can be inhibited by antibodies directed against MLV Env.

### Modulation of ERV expression affects NM aggregate induction

The foregoing experiments suggested that Env plays a prominent role in intercellular aggregate transmission and induction. We tested if silencing of MLV in donors affects aggregate induction in recipient cells. Transfection of donor cells with three individual siRNAs targeting MLV *env* or *gag/pol* (ORFs for *gag* and *pol* are overlapping) only slightly decreased MLV gene products, likely due to multicopy ERVs[19] (Fig. 3a–c). Partial knock-down of *gag/pol* or *env* still decreased NM aggregate induction in cocultured N2a NM-GFP cells (Fig. 3d), arguing that Env as well as Gag/Pol contribute to enhanced proteopathic seed spreading.

MLV expression underlies epigenetic control through promoter methylation early during development[32]. As DNA methyltransferase inhibitors can induce expression of ERVs[33], we tested if erasing this repressive epigenetic mark potentiates aggregate-inducing activity of

donor cells (Fig. 3e). Treatment of donor cells at early passage with DNA methyltransferase inhibitors 5-Azacytidine (Aza) or Decitabine (Dec) resulted in increased expression of MLV *env* and *gag* mRNA (Fig. 3f) and increased Env and Gag protein (Fig. 3g). Both drugs also significantly increased NM aggregate induction in recipient cells when these were cocultured with pretreated donors (Fig. 3h). In a reverse experiment, increased methylation by treatment of late passage donors with L-methionine, betaine or choline chloride resulted in decreased *env* and *gag/pol* mRNA (Fig. 3i), protein levels (Fig. 3j) and reduced aggregate induction in recipients (Fig. 3k). We conclude that modulation of MLV expression affects intercellular aggregate spreading in our N2a cell culture system.

### Viral protein maturation required for NM aggregate induction

MLV proteins Env and Gag require the MLV viral protease for processing into mature proteins (Supplementary Fig. 3a–d)[34,35]. To investigate if increased NM aggregate induction depends on the proper maturation of viral proteins, we tested anti-HIV-1 drugs for their effects on NM aggregate induction in coculture. Among the four tested HIV protease inhibitors, Amprenavir and Atazanavir have previously been shown to inhibit MLV protease[36]. Treatment of cocultures with these compounds (Fig. 4a) had no effect on the percentage of donor cells with pre-existing NM-HA aggregates (Supplementary Fig. 5a). However, the two MLV protease inhibitors Atazanavir and Amprenavir reduced the percentage of recipient cells with induced NM-GFP aggregates (Fig. 4b). Other HIV protease inhibitors, reverse transcriptase inhibitors or Hepatitis C virus (HCV) protease inhibitors had no effect on donor aggregates or aggregate induction in recipients during coculture (Supplementary Fig. 5b).

We hypothesized that Amprenavir inhibited MLV protease-driven viral protein maturation in donor cells or during EV formation and thus had no effect on recipients. Thus, we tested the effect of the HIV protease inhibitor Amprenavir on NM-GFP aggregate induction by treating recipient cells 1 h prior to the addition of EVs isolated from donor cells (LP) and incubated them together in the presence of the drug for a further 12 h. As a control, donor cells (LP) were also cocultured with recipients in the presence of the drug (Fig. 4c). As expected, Amprenavir only inhibited aggregate induction in cocultures, but not in drug-treated recipients exposed to EVs (Fig. 4d). By contrast, when donor cells (LP) were preincubated with different concentrations of Amprenavir for 3 days and subsequently cocultured with recipients in the presence of the inhibitor, aggregate induction in recipients was drastically decreased in a dose-dependent manner (Fig. 4e, f). Likewise, EVs isolated from Amprenavir-treated donors (LP) exhibited strongly reduced aggregate-inducing activity in recipient cells (Fig. 4g, h). Amprenavir did not affect secreted particle numbers (Fig. 4i), but impaired viral protein maturation (Supplementary Fig. 6). These experiments suggest that maturation of endogenous MLV-encoded gene products in donor cells or donor-derived EVs is required for efficient aggregate induction in recipient cells.

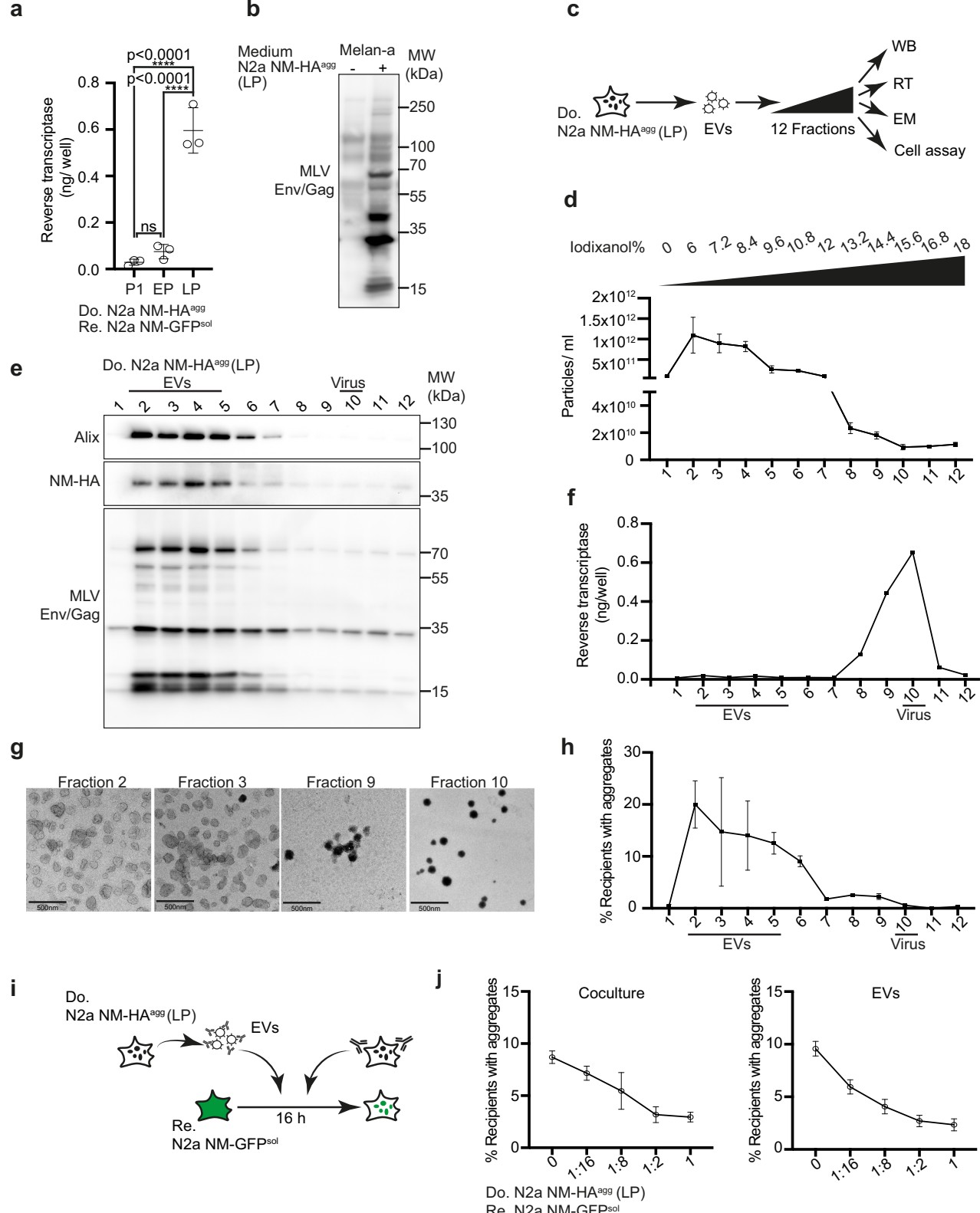

**Viral ligand–receptor interactions drive NM aggregate induction**

MLV envelope proteins mediate specific contact of virions with their cognate receptors on target cells and induce cargo release into the cytosol by enforcing the fusion of lipid bilayers. According to our proteomic analysis, polytropic Env was expressed in LP cells, suggesting that contact between polytropic Env and its receptor XPR1 was involved in NM aggregate spreading. Silencing of recipient MLV receptor XPR1 but not mCat-1, the receptor for ecotropic MLV (Fig. 5a–c), strongly reduced NM aggregate induction in cocultures (Fig. 5d) and by EV exposure (Fig. 5e), confirming that produced MLVs belong to the group X/P-MLVs using this receptor. Silencing of both receptors in recipient cells had no effect on NM aggregate induction by recombinant NM fibrils, arguing that NM aggregate uptake was

**Fig. 2 | Donor cells of late passage produce both EVs and active retroviral particles. a** Viral particles and EVs were precipitated from a conditioned medium of N2a NM-HA^agg cells of different passage numbers using polyethylene glycol. Reverse transcriptase (RT) activity was determined at P1, early passage (EP), and late passage (LP) using a colorimetric RT assay (Roche). **b** MLV-susceptible Melan-a cells were exposed to the conditioned medium of the donor clone. Western blot analysis was performed 6 days later using antibody ABIN457298 against Env/Gag. **c** The 100,000 × g pellet from the conditioned medium from donor clone N2a NM-HA^agg (late passage) was subjected to an Optiprep density gradient. Fractions were analyzed for particle numbers, reverse transcriptase (RT) activity, particle morphology by electron microscopy (EM), protein content by Western blot (WB), and aggregate-inducing activity. **d** Particle numbers of gradient fractions are determined using ZetaView. **e** Density gradient fractions were analyzed for Alix, endogenous Env/Gag (ABIN457298) and NM-HA. Alix and NM-HA were detected on the same blot, Env was detected on a separate blot by WB. **f** Reverse transcriptase (RT)

activity identifies viral particles (fractions 8–11). **g** Transmission electron microscopy of particles in fractions 2, 3, 9, and 10. Scale bar: 500 nm. **h** NM-GFP aggregate-inducing activity of gradient fractions in recipient cells. N2a NM-GFP^sol cells exposed to different fractions were analyzed for induced NM-GFP aggregates 16 h post-exposure. **i** Recipient cells were cocultured with donor cells (LP) in the presence of different dilutions of anti-MLV Env antibody mAb83A25 for 1 h. Alternatively, recipients were cultured with donor EVs that had been pre-incubated with anti-Env antibodies for 1 h. Anti-Env antibodies were present throughout the experiment. Partly created with Biorender.com. **j** The percentage of recipient cells with induced NM-GFP aggregates was determined 16 h post coculture or EV addition. All data are shown as the means ± SD from three (**a**, **d**, **f**, **h**), 12 (**j**, coculture) or 6 (**j**, EVs) replicate cell cultures. Three (**a**, **d**, **f**, **h**, **j**), two (**e**), or one (**g**) independent experiments were carried out with similar results. *P*-values were calculated by one-way ANOVA with Tukey's multiple comparisons. Source data are provided as a Source Data file.

mediated by EV–receptor contact and not the direct contact of an NM seed with the receptor (Supplementary Fig. 7a).

XPR1 is a receptor with eight putative transmembrane domains and four extracellular loops (ECL) (Fig. 5f). At least six polymorphic variants of XPR1 restrict infection by specific X/P-MLVs, with polymorphisms in ECL 3 and 4 affecting the entry of certain X/P-MLV subtypes[37]. Analysis of XPR1 of susceptible N2a NM-GFP^sol cells demonstrated that its Env recognition domain differed at 9 residues within ECL 3 and 4 from XPR1 expressed by HEK NM-GFP^sol cells (Fig. 5g; Supplementary Fig. 7b), a cell line refractory to NM aggregate induction by N2a NM-HA^agg-derived EVs. We generated a clonal HEK NM-GFP^sol cell line stably expressing an epitope-tagged N2a XPR1 variant (Supplementary Fig. 7c). Expression of the N2a XPR1 variant (Fig. 5h) conferred susceptibility to HEK cells, both in coculture or upon exposure to donor-derived EVs (Fig. 5i, j). By contrast, expression of the N2a polymorphic XPR1 variant had no effect on aggregate induction by recombinant NM fibrils (Supplementary Fig. 7d). We conclude that efficient NM aggregate induction via coculture or EVs depends on the specific interaction of Env with its receptor. As xenotropic MLVs are unable to infect murine cells[38], the respective Env involved in aggregate spreading is of polytropic origin.

### Retroviral proteins promote the spreading of NM and Tau aggregates

The foregoing experiments demonstrated that the upregulation of MLV strongly affected intercellular aggregate spreading. So far it was unclear if generation of active viral particles was required for this process. To test this, HEK NM-HA^agg cells not coding for MLV were transfected with combinations of plasmids coding for MLV *gag/pol*, amphotropic MLV *env* 10A1 and MLV transfer vector for virus production (Fig. 6a). Vero cells stably expressing NM-GFP^sol[7] were chosen as recipients due to their high expression level of amphotropic Env receptor Pit2 (Fig. 6b). Ectopic expression of viral protein Env resulted in significantly increased aggregate induction rates in cocultured recipients, with higher induction rates when Env and Gag/Pol were expressed simultaneously (Fig. 6c, d). Highest induction rates in cocultures were observed when donor cells were also transfected with retroviral transfer vector (TV) (Fig. 6e). We confirmed that cells transfected with all vectors produced active viral particles by exposing wildtype Vero cells to conditioned medium from donors transfected with packaging plasmids and TV coding for Luciferase (Fig. 6f, g). EV concentrations in the conditioned medium increased when cells were transfected with an *env* and a *gag/pol* construct (Fig. 6h). Highest vesicle numbers were observed when cells were also transfected with TV, thus producing virus (Fig. 6h). When adjusted for comparable vesicle numbers, conditioned medium of cells producing active virus also most efficiently induced protein aggregation in recipients (Fig. 6i).

Next, we tested if retroviral proteins also promoted intercellular transmission of protein aggregates associated with neurodegenerative

diseases. To this end, we made use of our recently developed cell culture model propagating aggregates composed of the repeat domain of a mutant human Tau protein (Fig. 6j)[39]. HEK cells stably expressing a soluble GFP-tagged repeat domain variant of human Tau (hereafter termed Tau-GFP^sol) stably produce and maintain Tau-GFP aggregates upon exposure to AD brain homogenate (Tau-GFP^AD)[7,39]. Upon donor transfection with retroviral plasmids, induction rates in cocultured Vero cells expressing the same Tau variant fused to FusionRed (Tau-FR^sol)[7] were significantly increased (Fig. 6k–n). Again, highest induction rates were observed when donors were transfected with plasmid coding for *gag/pol* and *env* and TV (Fig. 6n). Transfection of combinations of *gag/pol* and *env* vectors also strongly increased vesicle release (Fig. 6o). When conditioned medium was adjusted for comparable particle numbers, highest induction rates were observed when donors were transfected with both MLV *gag/pol* and *env* constructs (Fig. 6p). We conclude that expression of retroviral proteins Env, Gag and Pol is sufficient to promote intercellular proteopathic seed spreading, but that spreading is most efficient when donor cells produce active viral particles.

### HERV Envelope proteins increase proteopathic seed spreading

In contrast to murine endogenous retroviruses of the MLV clade, HERVs have so far not been shown to produce infectious virions in vivo. However, under certain circumstances, HERVs become reactivated, resulting in transcript and even protein expression. Upregulation of different HERV Env proteins, including HERV-W and HERV-K, has been observed in several neurological or neurodegenerative diseases[9,16,17]. Of the hundreds of HERV-W elements that exist in the human genome, several contain full or partial proviral structures. An element within this group, ERVW1, encodes an intact open-reading frame for Env, the so-called Syncytin-1. Syncytin-1 has fusogenic activity and has been co-opted as a cellular gene for placenta formation. The most recently acquired HERV-K subtype HML-2 is present in hundreds of copies in the human genome, with approx. 90 copies encoding functional proteins[40]. While the entry receptor for HERV-K HML-2 K113 is unknown, a consensus sequence for the HERV-K HML-2 K113 has previously been shown to mediate the uptake and replication of pseudotyped viruses in a broad range of mammalian cells, including HEK cells[41].

To assess if Env proteins encoded by HERV clades implicated in neurological diseases affect spreading of proteopathic seeds, donor HEK NM-HA^agg cells were transfected with plasmids coding for epitope-tagged HERV-W Env Syncytin-1 or coding for an Env consensus sequence of the HERV-K HML-2 subgroup K113 (from now on termed HERV-K Env)[41] (Fig. 7a, b). Empty vector served as a control. Coculture experiments with HEK cells expressing NM-GFP^sol revealed that both HERV Envs resulted in a significant increase in recipient cells with NM-GFP aggregates (Fig. 7c, d). Likewise, expression of HERV Envs also increased intercellular Tau aggregate spreading in HEK cell cocultures

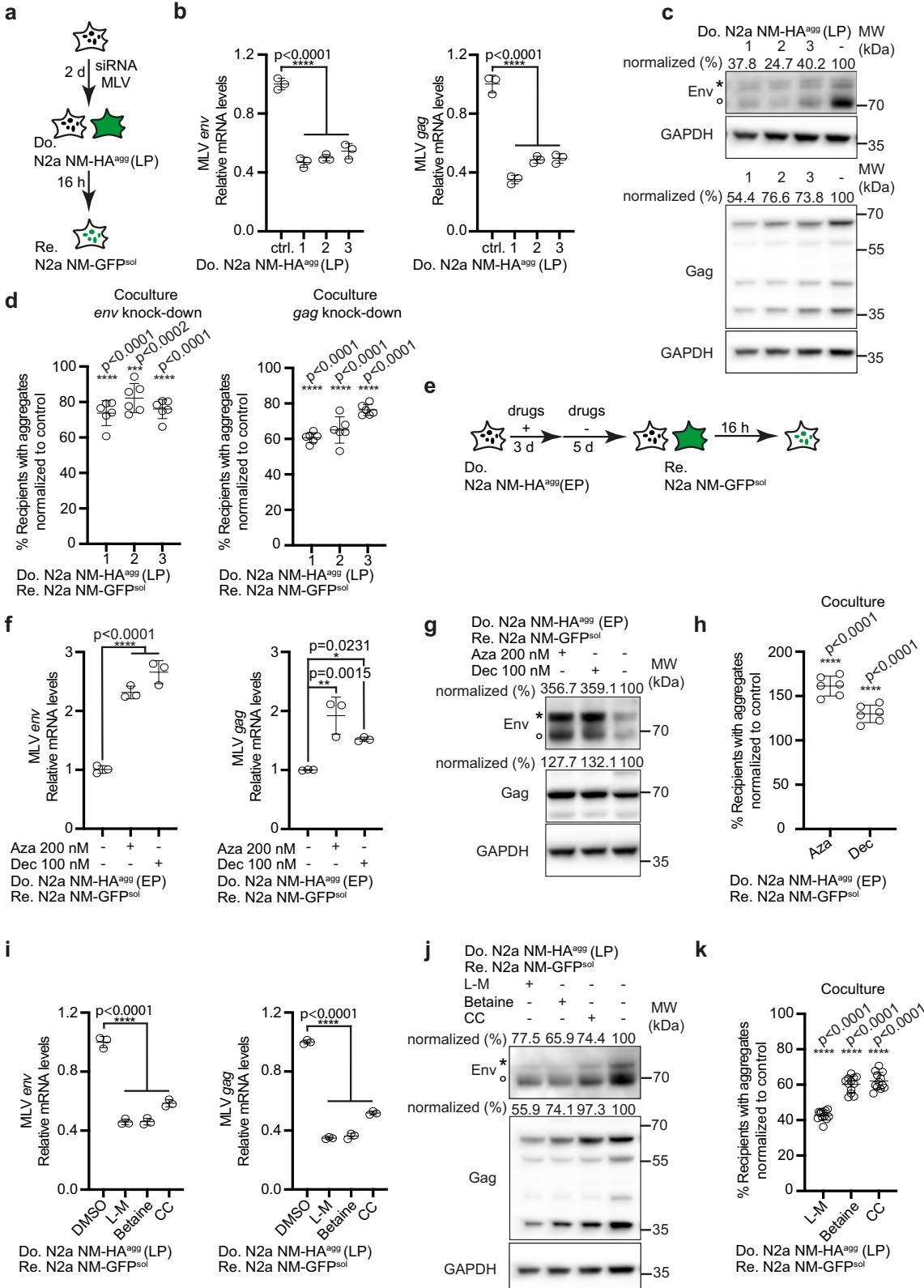

(Fig. 7e-h). No Tau-FR$^{sol}$ aggregation was observed when HEK Tau-FR$^{sol}$ cells were cocultured with Env-expressing donors that lacked Tau-GFP aggregates, arguing that HERV Env expression did not trigger spontaneous Tau aggregation (Supplementary Fig. 8a, b). The interaction of HERV-W Syncytin-1 with its cognate receptors ASCT1/2[42] can be inhibited by feline ERV RD114 Env which binds to and thereby blocks the same receptors[43]. Expression of ERV RD114 Env in recipients

abolished the HERV-W Syncytin-1 mediated increase in intercellular aggregate induction (Supplementary Fig. 8c, d). Truncation of the carboxyterminal domain has been demonstrated to increase the specific entry and fusogenic properties of Syncytin-1[44]. Truncated Syncytin-1 further increased aggregate induction in bystander cells (Supplementary Fig. 8e–g). Similar to HERV-W Syncytin-1, carboxyterminally deleted HERV-K Env shown to facilitate efficient entry into

**Fig. 3 | Epigenetic modulation of MLV expression affects aggregate spreading.**
**a** Donor N2a NM-HA[agg] cells of late passage (LP) were transfected with three siRNAs against endogenous *env*, *gag*, or non-silencing siRNA as control and subsequently cocultured with recipients. **b** Reduction of donor transcripts following siRNA treatment assessed by qRT-PCR. Shown are fold changes relative to mock control. **c** Env and Gag expression in N2a NM-HA[agg] cells (LP) transfected with siRNA. Note that viral transcripts include a genomic transcript coding for *gag/pol* and *env* and a spliced transcript coding only for *env*. Antibodies were anti-MLV Env mAb83A25, anti-MLV Gag ab100970 and anti-GAPDH. Env and Gag Western blots were reprobed for GAPDH. Asterisk marks the uncleaved Env precursor, circle marks the SU domain. Modulation of Env expression is expected to change the rate of newly translated precursor to processed Env. Protein levels normalized to GAPDH are shown. **d** Donors cocultured with recipients 72 h post-transfection. Shown is the percentage of recipient cells with NM-GFP[agg] normalized to control**. e** Donors (EP) were treated with 5-azacytidine (Aza), 5-aza-2′-deoxycytidine (Dec), or DMSO for 3 days and then cultivated in a normal medium for 5 days. Cells were subsequently cocultured with recipients for 16 h. **f** Expression of MLV *env* (left panel) or *gag* (right panel) transcripts in donor cells 5 days post-treatment assessed by qRT-PCR. **g** Western blot of Env and Gag after inhibitor treatment. The same blot was probed with anti-MLV Env mAb83A25, anti-MLV Gag ab100970 and anti-GAPDH. **h** Percentage of NM-GFP[agg] recipient cells. **i** Donors (LP) were treated with L-methionine (L-M), betaine, choline chloride (CC), or medium control for 6 days. MLV *env* or *gag* transcripts were analyzed by qRT-PCR. **j** Expression of Env, Gag, and GAPDH probed on the same blot. **k** Percentage of NM-GFP[agg] cells. All data are shown as the means ± SD from three (**b**, **f**, **i**), six (**d**, **h**), or 12 (**k**) replicate cell cultures. Three (**b**, **d**, **f**, **h**, **i**, **k**) independent experiments were carried out with similar results. *P*-values were calculated by one-way ANOVA with Dunnett's post hoc test. Source data are provided as a Source Data file.

susceptible host cells[41] also increased intercellular aggregate induction, arguing that viral ligand–receptor interactions and virally induced lipid bilayer merging drive intercellular protein aggregate dissemination, as we have shown for exogenous viral envelope proteins[7] (Supplementary Fig. 8h–j).

To assess if the effect of HERV Env was only observed for HERV-K and HERV-W clades, envelope proteins of HERV clades R and H were similarly tested for their effect on protein aggregate spreading. Again, enhanced intercellular protein aggregate spreading was observed, suggesting that reactivation of diverse HERV clades could contribute to the spreading of protein misfolding (Supplementary Fig. 9a–h). No spontaneous aggregate induction was observed when donors expressing soluble Tau-GFP[AD] were used in cocultures (Supplementary Fig. 9i, j). Real-time PCR demonstrated that the expression of inflammatory cytokines was largely unaffected by the expression of HERV Envs in HEK Tau-GFP[AD] donors, arguing that these did not play a major role in intercellular aggregate dissemination in our cell-based assay (Supplementary Fig. 9k).

Lastly, we tested if HERV-W or HERV-K Env expression by HEK donors also increased aggregate induction in primary human astrocytes. While entry receptors for HERV-W are expressed by neurons and glia[45], the receptor for HERV-K is unknown. Importantly, HERV-K and -W Env expression in HEK Tau-GFP[AD] donors also increased Tau-FR aggregation in primary human astrocytes, demonstrating that this effect was not restricted to immortalized cells (Fig. 7i–k). We conclude that, similar to murine endogenous retroviruses, the expression of HERV gene products can increase the intercellular spreading of diverse proteopathic seeds.

## Discussion

Accumulating evidence argues that HERVs, resulting from retroviral germline invasions throughout evolution, are upregulated in NDs. Some HERV gene products can be directly neurotoxic, such as HERV-W Env[46] and HERV-K Env[47]. Further, inflammatory responses due to viral transcripts have been implicated in ND development[14]. Our data suggest an additional mechanism by which ERV proteins can contribute to neurodegeneration, namely by accelerating intercellular dissemination of protein particles. Activation of polytropic endogenous MLV proviruses resulted in the production of infectious virions as well as the secretion of protein aggregate-loaded EVs decorated with viral Env. As a result, MLV upregulation drastically increased the intercellular transmission of proteopathic seeds by EVs to bystander cells or to cells in direct contact and induced protein aggregation in the latter.

The large number of MLV proviruses that populate the murine genome are usually replication incompetent and/or epigenetically silenced[20]. Recombination between derepressed MLV subgroups results in the formation of infectious intersubgroup variants with a polytropic host range[48]. Our data suggest that the emergence of active viruses in our cellular system is likely due to the reactivation of silenced MLV proviruses followed by viral recombination. The effect of

epigenetic drugs on MLV expression suggests that the demethylation of MLV promoter regions contributes to this phenomenon. The activation of ERVs was independent of transgene expression or the induction of protein aggregates. Activation and generation of ERV-derived retroviral particles have been reported for several cell lines in culture, including N2a cells, but the cellular processes that initiate the derepression of proviruses remain unknown[26,27,49,50]. Interestingly, receptor usage of viral particles produced in our N2a cell model clearly differed from the ecotropic MLV identified by others, arguing that within a given cell population, different MLVs can become produced[27,49].

Our detailed analysis reveals that the effect of activated endogenous MLV on protein aggregate spreading can be attributed to the expression of Env glycoprotein and retroviral Gag/Pol polyproteins. Upon increased expression, Env on the cell surface or on EVs mediates the binding to specific receptors on the cell surface of recipient cells. Cleavage of the MLV Env R-peptide by MLV protease then initiates the fusion of cell membranes or EVs with the recipient cell or its endolysosomal membranes, resulting in the release of proteopathic seeds into the cytosol of the recipient cell and subsequent aggregate induction. Surprisingly, reconstitution experiments in HEK cells demonstrate that MLV Env alone is sufficient to increase intercellular aggregate spreading, in analogy with our findings that vesicular stomatitis virus G and SARS-CoV2 spike S glycoproteins can elevate intercellular aggregate induction[7]. The additional positive effect of co-transfecting a plasmid coding for Gag and Pol polyproteins on aggregate induction is likely due to more efficient R peptide cleavage as well as the concentration of Env and Gag polyproteins within rafts[51]. However, the highest induction rates were achieved when all plasmids for MLV production were transfected into donors. Thus, cells harboring protein aggregates and also actively producing infectious MLV most efficiently induce proteopathic seeds in bystanders. The reason for this is unclear but might be related to the fact that MLV RNA increases the efficiency of proper Gag–Gag interactions, which in turn also affect proper Env positioning within rafts[52].

Most experiments in this study were performed using a model protein for cytosolic protein aggregates, which is based on the prion domain of the *S. cerevisiae* translation termination factor Sup35. The prion domain of Sup35 shares striking compositional similarity with so-called prion-like domains of a growing number of proteins associated with ALS and FTLD, such as FUS and TDP-43[22]. In analogy to our study, experiments with transgenic mice have recently demonstrated that misfolded aggregated TDP-43 is secreted within EVs, suggesting that protein aggregates with similar domains are secreted by the same mechanisms[53]. Further, EVs isolated from the plasma of ALS patients are enriched for TDP-43 and FUS[54]. Thus, it is feasible to assume that other protein aggregates with prion-like domains might also be affected by ERV derepression. This hypothesis is supported by our findings that retroviral gene expression in donor cells harboring Tau aggregates also increased aggregate dissemination, demonstrating

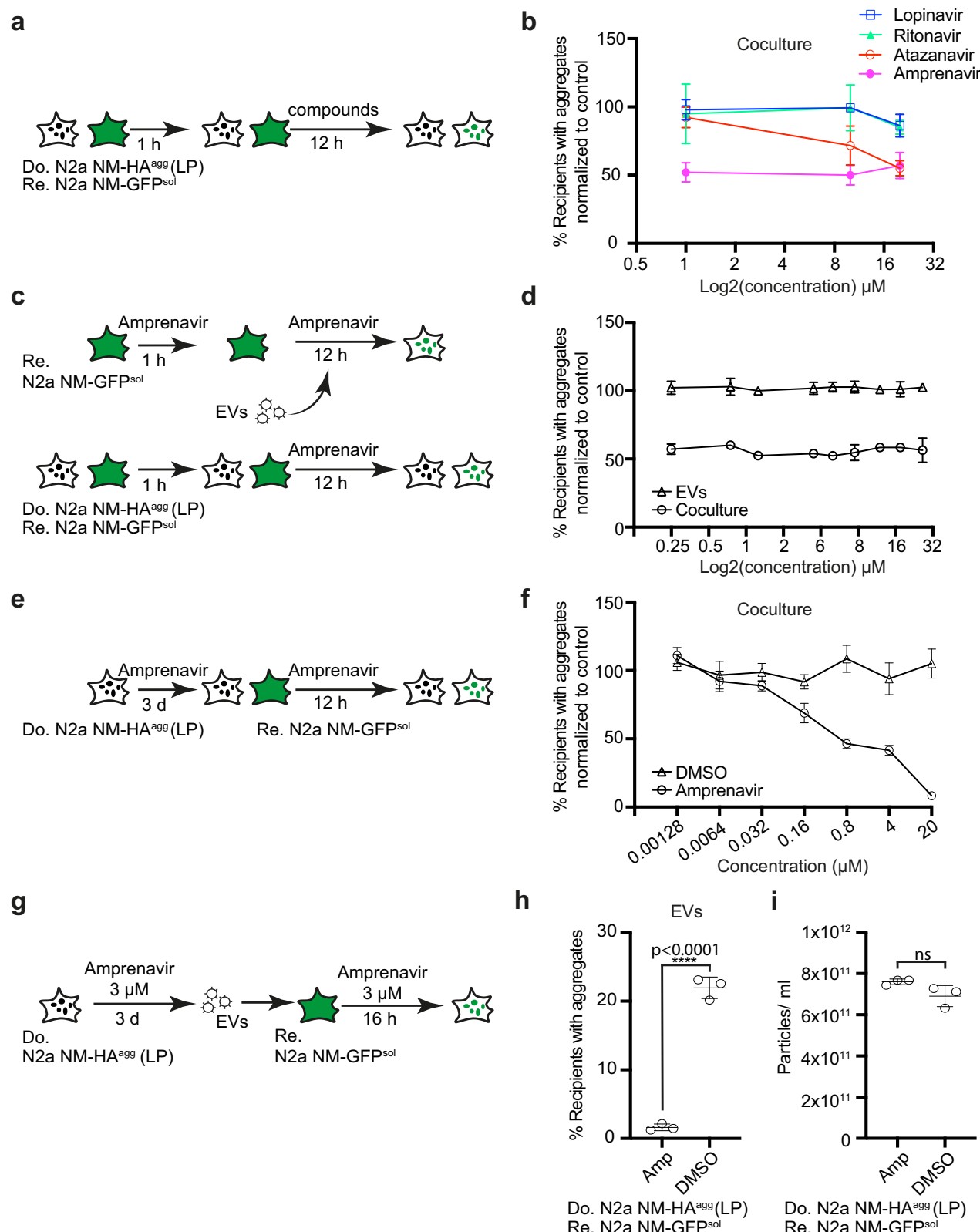

that this effect was independent of the type of cytosolic protein aggregate. Our results are consistent with a previous study, showing that simultaneous infection of cells with scrapie agent and friend retrovirus strongly enhances intercellular spreading of pathologic prion protein and scrapie infectivity[55]. Viral Env and Gag association with prion-containing EV fractions has previously been observed for a prion-infected, endogenous ecotropic MLV-producing N2a

subpopulation. However, the role of Env expression in intercellular prion spreading had not been studied[27]. Interestingly, independent in vivo co-infections with MLV and scrapie showed no effect on scrapie incubation times, potentially because target cells for exogenous MLV and scrapie differ[56]. Unfortunately, the effect of derepressed MLVs on the spreading of proteopathic seeds cannot easily be tested in ND mouse models, as endogenous MLV proviruses in common mouse

**Fig. 4 | Protease inhibitors blocking MLV maturation impair intercellular protein aggregate spreading. a** Workflow of compound tests in cocultures. Donor clone N2a NM-HA^agg (LP) and recipient N2a NM-GFP^sol cells were co-seeded and viral protease inhibitors were added at three concentrations 1 h later. 12 h post-drug treatment, donor and recipient cells were analyzed for the percentage of donor cells with NM-HA^agg or recipient cells with induced NM-GFP^agg. **b** Quantitative analysis of the effect of protease inhibitors on cells with NM-GFP aggregates. Recipients with NM-GFP aggregates that were solvent-treated (DMSO) were set to 100%. **c** Workflow of compound test in cocultures. Donor N2a NM-HA^agg (LP) and recipient N2a NM-GFP^sol were co-seeded and exposed to different concentrations of Amprenavir 1 h later. Alternatively, recipient N2a NM-GFP^sol cells were pretreated with different concentrations of Amprenavir for 1 h, and cells were subsequently exposed to donor-derived EVs for 12 h. Note that donor cells (LP) from which EVs were isolated remained untreated. **d** Amprenavir effects on the percentage of recipient cells with induced NM-GFP aggregates in the two assays. Results were normalized to solvent control. **e** Donor N2a NM-HA^agg cells (LP) were treated with different concentrations of Amprenavir for 3 days. Cells were subsequently cocultured with recipient N2a NM-GFP^sol. Aggregate formation in recipients was assessed 16 h later. **f** Dose-dependent inhibition of NM-GFP aggregation in cocultured recipient cells upon Amprenavir treatment. Data were normalized to DMSO-treated control set to 100%. **g** EVs were harvested from treated donors (LP) and added to recipient cells. **h** Induction of NM-GFP aggregates in recipient cells by EVs derived from Amprenavir-treated donor cells (LP). **i** Effect of Amprenavir on particle release by donor cells (LP). All data are shown as the means ± SD from two (**d**), three (**b, h, i**), or six (**f**) replicate cell cultures. Three (**b, d, f, h, i**) independent experiments were carried out with similar results. *P*-values calculated by two-tailed unpaired Student's *t*-test (**h, i**). ns: non-significant. Source data are provided as a Source Data file.

lines such as C57BL/6 used in ND research are transcribed at low or undetectable levels[57]. Endogenous polytropic MLV proviruses are highly polymorphic, but stable DNA elements are widespread in murine genomes[19]. Under certain circumstances such as impaired immunity[21], endogenous MLV can produce infectious virions, a characteristic that differs from HERVs which are considered non-infectious. Still, also HERVs have been shown to produce viral-like particles in cell culture, cancer, and autoimmune diseases[58,59]. For example, RNA and protein transcribed from LINE-1 retroelements are packaged into EVs[60]. HERV-W Syncytin-1 is present on EVs derived from placenta[61].

Accumulating evidence suggests an association of aberrant HERV expression with NDs. Some but not all studies demonstrate that HERV-K transcripts are more abundant in ALS patients compared to unaffected individuals[17]. The expression appears to be restricted to neurons rather than glia[47]. Elevated levels of HERV-K Env peptides in sera and CSF of ALS patients correlated with poor functional performance, suggesting that HERV-K Env contributes to disease progression[62,63]. HERV-K Env was also found on neuronal EVs isolated from the plasma of patients suffering from motor neuron disease[64]. Impaired ERV repression has also been correlated with Tau pathology[11,12]. Elevated ERV transcripts have been reported for AD[13,14], PSP[15], behavioral variant frontotemporal dementia[16] and sporadic Creutzfeldt-Jakob disease[65]. Interestingly, HERV-W Env variants expressed in multiple sclerosis patients can also be secreted as soluble hexamers[66]. Their role in other neurodegenerative diseases such as AD remains to be established. Expression of retroelements has also been observed in TSEs. Brains of BSE-infected macaques displayed increased ERV transcripts[67]. ERV Gag protein and RNA of the retroelement group IAP were also found to co-fractionate with CJD infectivity[68]. Importantly, HERVs are also upregulated during infection with exogenous pathogens, during inflammation and aging, processes which have been implicated in the progression of NDs[69]. While evidence exists that HERV transcripts or gene products can be directly neurotoxic or induce neuroinflammation, we here show that ERV activation could also contribute to prion-like spreading events. Interfering with ERV gene product expression and maturation could thus be a promising strategy for disease intervention.

## Methods
### Ethics statement
SWISS mice were housed and handled according to standards of the German Animal Welfare Act under standard conditions (23 °C, 40–50% humidity, ad libitum access to food and water) with a 12 h light/dark cycle. Mice were euthanized according to the German Animal Welfare Act for organ harvest according to Section 4 Abs. 3 TierSchG. Under German law, organ harvest is notifiable but not subject to approval. Authorities were notified of the work.

### Plasmids
Murine XPR1 or human XPR1 were amplified from cDNA of N2a NM-GFP^sol or HEK NM-GFP^sol cells. The open-reading frame of murine XPR1 was tagged with a sequence for the HA-epitope and cloned into PiggyBac expression vector PB510B-1 (System Biosciences) with XbaI and NotI. To generate the luciferase reporter gene plasmids, the respective open-reading was tagged with a sequence for the V5-epitope and cloned into pHCMV-EcoENV (Addgene #15802) using EcoRI and XhoI to replace EcoENV. For retroviral transduction, BamHI and EcoRI were used to clone the open-reading frame into the retroviral transfer vector pSFF. To generate the pHCMV-Syncytin-1-100UTR plasmid, Syncytin-1 cDNA tagged with a V5 epitope sequence (#T0264; GeneCopoeia) was cloned into pHCMV-EcoENV (Addgene #15802) using EcoRI and XhoI to replace EcoENV. The consensus sequence for Env encoded by the HERV-K HML-2 K113 element has previously been published[41]. The codon usage optimized sequence with a V5 tag was synthesized (Biocat) and cloned into pHCMV via EcoRI and XhoI. For the generation of HERV-H and HERV-R Env constructs, codon usage optimized sequences with V5 tags were synthesized (Biocat) and cloned into pHCMV via EcoRI and XhoI based on a consensus sequence for HERV-H Env (GenBank accession CAB94193.2; Q9N2K0.1; AAD34324.1) and HERV-R Env (GenBank accession AAA88027.1). HERV-KΔ construct was cloned according to Kramer et al.[70] and Syncytin-1Δ was cloned according to Uygur et al.[71]. The 100 bp sequence from 3´-UTR of Syncytin-1 shown to enhance gene expression[72] was amplified using primers (forward: 5´-CCGCTCGAGAGCGGTCGTCGGCCAAC-3´/ reverse: 5´-GAAGATCTCCTTCCCAGCTAGGCTTAGGG-3´) and genomic DNA from MCF-7 cells as template. The sequence was cloned into pHCMV-Syncytin-1 using XhoI and BglII restriction sites. pHCMV-10A1 was a gift from Miguel Sena-Esteves (Addgene plasmid # 15805)[73] and pBS-CMV-gagpol was a gift from Patrick Salmon (Addgene plasmid # 35614). pLTR-RD114A was a gift from Jakob Reiser (Addgene plasmid # 17576).

### Cell lines
N2a and HEK 293T cells expressing NM-HA^sol or NM-GFP^sol and N2a and HEK NM-HA^agg clones induced to propagate NM-HA aggregates have been published previously[23,24]. HEK NM-GFP^sol cells stably expressing the HA-tagged N2a XPR1 variant were generated using the Piggyback transposon system (System Biosciences) and transposase followed by clonal selection with puromycin. Vero NM-GFP^sol, Vero Tau-FR^sol, HEK Tau-GFP^sol, and HEK Tau-FR^sol cells have been described previously, as have HEK Tau-GFP^AD cells producing Tau-GFP aggregates induced with AD brain homogenate[7,39]. All code for human 4R Tau amino acids 243–375 with mutations P301L and V337M fused to GFP or FR (Evrogen) with an 18-amino acid flexible linker (EFCSRRYRGPGIHRSPTA), as described previously[74]. N2a and HEK cells were cultured in

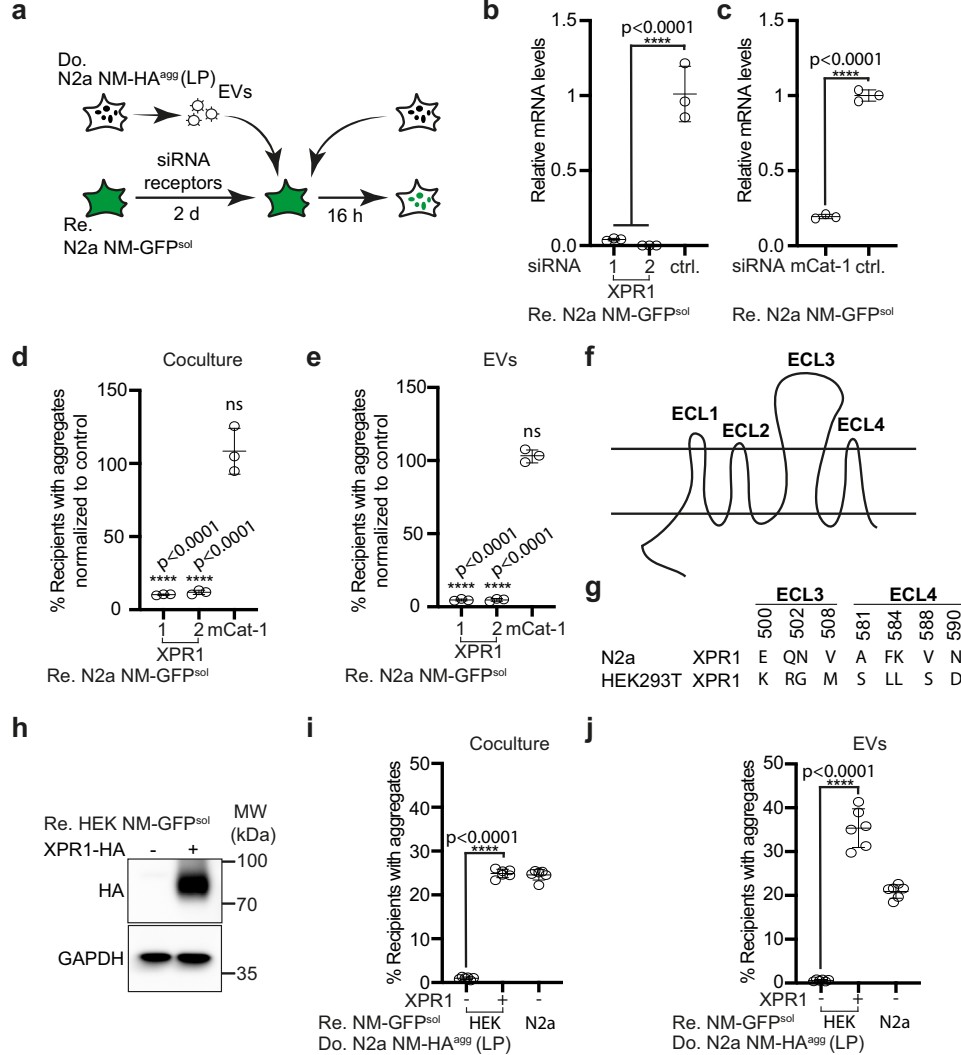

**Fig. 5 | Receptor polymorphisms modulate intercellular proteopathic seed spreading. a** Experimental workflow. Recipient N2a NM-GFP$^{sol}$ cells were transfected with two siRNAs against XPR1, one mCat-1 siRNA, or a non-silencing siRNA control. 48 h later, recipient cells were cocultured with donor cells (LP). Alternatively, recipients were exposed to EVs isolated from conditioned medium of N2a NM-HA$^{agg}$ cells (LP). NM-GFP aggregate induction was determined 16 h post-exposure or coculture. **b** Knock-down of XPR1 mRNA by two independent siRNAs was assessed 48 h post-siRNA transfection by qRT-PCR. Shown is the fold change of mRNA expression normalized to control (ctrl.). **c** Knock-down of mCat-1 mRNA. **d, e** Recipients with NM-GFP aggregates following receptor knock-down. Shown are the results of coculture (**d**) and of recipients exposed to donor-derived EVs (LP) (**e**). NM-GFP aggregate induction was measured 16 h post EV addition or coculture. **f** Transmembrane structure of XPR1. The receptor contains four extracellular loops (ECL1–4)[84]. **g** Polymorphic variants of xenotropic and polytropic X/P-MLV receptor

XPR1 in mouse N2a and human HEK cells. Shown are mismatches in the surface-exposed loops ECL 3 and 4. ECL 3 and 4 are required for the binding of X/P-MLV[84]. **h** Ectopic expression of the N2a XPR1 receptor variant in poorly permissive HEK NM-GFP$^{sol}$ cells. Ectopically expressed XPR1-HA was detected using anti-HA antibodies. GAPDH served as a loading control on the same blot. **i** HEK NM-GFP$^{sol}$ cells were transfected with mouse XPR1-HA or mock-transfected and subsequently cocultured with donor N2a NM-HA$^{agg}$ cells (LP). N2a NM-GFP$^{sol}$ cells served as recipient controls. **j** Alternatively, transfected HEK NM-GFP$^{sol}$ cells were exposed to EVs from N2a NM-HA$^{agg}$ cells (LP). N2a NM-GFP$^{sol}$ cells served as recipient controls. All data are shown as the means ± SD from three (**b**–**e**) or six (**i**, **j**) replicate cell cultures. Three (**b**–**e**, **i**, **j**) independent experiments were carried out with similar results. *P*-values calculated by two-tailed unpaired Student's *t*-test (**c**), one-way ANOVA with Dunnett's post hoc test (**b**, **d**, **e**), or one-way ANOVA with Bonferroni multiple comparisons (**i**, **j**). Source data are provided as a Source Data file.

Opti-MEM (Gibco) supplemented with glutamine, 10% (v/v) fetal bovine serum (FCS) (PAN-Biotech GmbH), and antibiotics. Vero cells were purchased from the cell lines service (CLS) and cultivated as recommended. Melan-a cells (Wellcome Trust, UK) were cultured in RPMI 1640 (Gibco) with 2 mM glutamine, 10% FCS, 1% penicillin/streptomycin, and 200 nM PMA. Human primary astrocytes (ScienCell, # 1800) were cultivated as recommended by the supplier. All cells were incubated at 37 °C and 5% $CO_2$. The total number and the viability of cells were determined using the Vi-VELL$^{TM}$XR Cell Viability Analyzer (Beckman Coulter). Transfections of cells were performed either with Lipofectamine 2000 (Thermo

Fisher Scientific, USA) or TransIT-2020/X2 (Mirus) reagents as recommended by the manufacturers.

### Isolation of cortical neurons

SWISS mice were obtained from Janvier-Labs (France). Preparation of cortical neurons was performed from the cortices of p13 pups according to the German AnimalWelfare Act, paragraph 4.3, and was approved by the DZNE animal welfare officer. Sex was not considered for the isolation of cortical neurons. Cortices with removed meninges were cut and cells were dissociated with 0.25% trypsin for 10 min at 37 °C. Tissue was passed three times through a 20 G needle. Rinsed

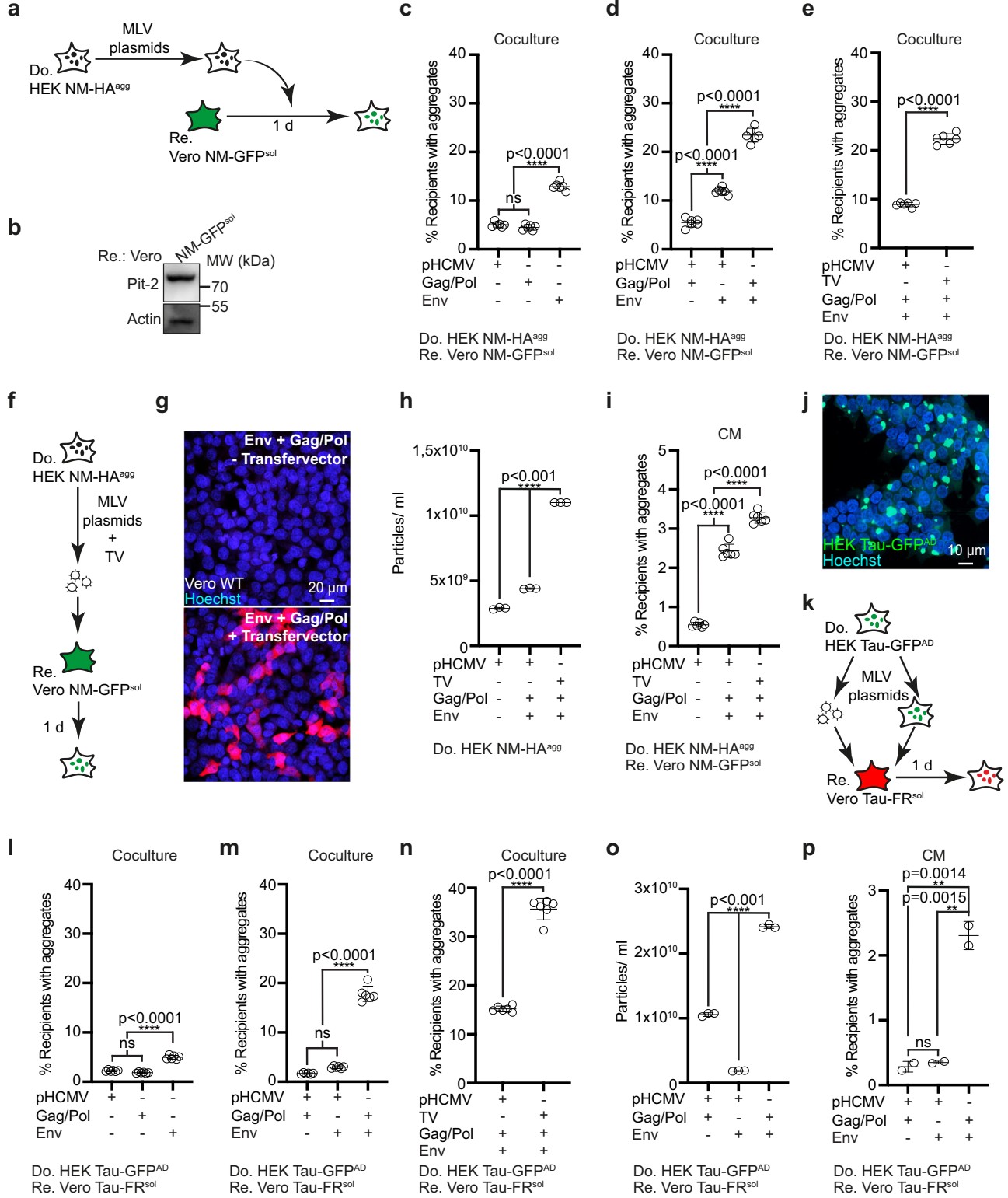

cells were filtered through a 100-µm cell strainer before plating them on 96-well plates or Sarstedt 8 slice chambers in Neurobasal medium with 2% B-27 supplement (Invitrogen), 1% L-glutamine (Invitrogen) and 1% penicillin/streptomycin (Invitrogen). All cell culture plasticware was pre-coated with 250 µg/mL poly-L-lysine (molecular weight >300 kDa, Sigma). Cortical neurons were transduced with lentivirus after 2 days of cultivation. 2 days later, EVs were added to the cortical neurons and incubated for 2 days. The neurons were fixed for microscopy and image analysis.

**Production and transduction with lentiviral particles**
HEK293T cells were cotransfected with plasmids pRSV-Rev (Addgene plasmid #12253), pMD2.VSV-G (Addgene plasmid #12259), pMDl.g/pRRE (Addgene plasmid #12251), and pRRl.sin.PPT.hCMV.Wpre containing Tau-FR or NM-GFP. Supernatants were harvested and concentrated with PEG. Vero cells, primary murine neurons, and human astrocytes were transduced with lentivirus. Stable Vero cell clones expressing Tau-GFP/FR were produced by limiting dilution cloning.

**Fig. 6 | MLV reconstitution increases intercellular NM and Tau aggregate induction. a** HEK NM-HA[agg] donor cells transfected with combinations of amphotropic MLV *env* 10A1 vector, *gag/pol* plasmid, retroviral transfer vector, and/or empty vector pHCMV were cocultured with recipient Vero NM-GFP[sol] cells. **b** Western blot of Vero NM-GFP[sol] cells expressing Pit-2. GAPDH was detected on the same membrane. **c** Recipient Vero cells with NM-GFP cocultured with donors transfected with single plasmids. **d** Recipients cocultured with donors transfected with combinations of plasmids. **e** Recipient cells with induced NM-GFP[agg] cocultured with donors that were additionally transfected with transfer vector (TV) for virus production. **f** Donors transfected with viral plasmids with TV coding for Luciferase-V5. Virus production was confirmed by transducing wild-type Vero cells with a conditioned donor medium. **g** Donors transfected with/without Luciferase-V5 coding retroviral TV and plasmids coding for *gag/pol* and *env* produce a virus that is infectious to wild-type cells, as demonstrated by immunofluorescence with anti-V5 antibodies. **h** Vesicle secretion upon transfection of viral plasmids. **i** Induction of NM-GFP aggregates in recipients exposed to conditioned medium (CM). **j** HEK Tau-GFP[AD] donor cell population. **k** HEK Tau-GFP[AD] donor cells transfected with combinations of plasmids coding for *env* 10A1, *gag/pol*, retroviral TV, and/or non-viral empty vector. Donors were cocultured with recipient Vero Tau-FR[sol] cells. Alternatively, donor medium was added to recipient cells. **l** Recipients with Tau-FR[agg] upon coculture with donors transfected with individual plasmids. **m** Recipients with induced Tau-FR[agg] upon coculture with donors transfected with plasmid combinations. **n** Recipients with induced Tau-FR[agg] upon coculture with donors transfected with plasmid combinations and TV. **o** Particle secretion upon transfection. **p** Recipients with Tau-FR[agg] exposed to conditioned medium adjusted for comparable particle numbers. All data are shown as the means ± SD from two (**p**), three (**h**, **o**) or six (**c–e**, **i**, **l**, **m**, **n**) replicate cultures. Three (**c–e**, **h**, **i**, **l–p**) independent experiments were carried out with similar results. *P*-values calculated by two-tailed unpaired Student's *t*-test (**e**, **n**) or one-way ANOVA with Tukey's multiple comparisons (**c**, **d**, **h**, **i**, **l**, **m**, **o**, **p**). ns: non-significant. Source data are provided as a Source Data file.

## EV isolation

To prepare the EV-depleted medium, FCS was ultracentrifuged at $100,000 \times g$, 4 °C for 20 h. Medium supplemented with EV-depleted FCS and antibiotics was subsequently filtered through a 0.22 and a 0.1 μM filter-sterilization device (Millipore). For EV isolation, $2–4 \times 10^6$ cells were seeded in a T175 flask in 35 ml EV-depleted medium to be confluent after 3 days.

EVs were harvested 3 days post-transfection. Cells and cell debris were pelleted by differential centrifugation ($300 \times g$, 10 min; $2000 \times g$, 20 min; $16,000 \times g$, 30 min), and the remaining supernatant (conditioned medium) was subjected to ultracentrifugation (UC) ($100,000 \times g$, 1 h) using rotors SW45Ti or SW32Ti (Beckman Coulter), before the pellet was rinsed with PBS and spun again using rotor SW55Ti ($100,000 \times g$, 1 h). For conditioned medium experiments, cells were transfected with plasmids pHCMV-10A1 (Addgene #15805), pBS-CMV-gag-pol (Addgene #35614), and pHCMV using TransIT. 5 h post-transfection, the medium was switched to medium with EV-depleted serum. The EV-conditioned medium was harvested 3 days post-transfection, centrifuged for 10 min at $300 \times g$, and subsequently used for aggregate induction assays.

## Aggregate induction assay

Recipient cells were cultured on CellCarrier-96 or 384 black microplates (PerkinElmer) at appropriate cell numbers for 1 h, and then treated with 5–20 μl of prepared samples (isolated EVs, conditioned medium, or recombinant NM fibrils). To destroy EVs, samples were sonicated for 5 min at 100% intensity (Sonicator Sonoplus, Bandelin). For cocultures, the recipient and donor cells were seeded at different ratios (donor: recipient 5:1). The total number of cells per 384-well was approx. $3 \times 10^4$. After an additional 18 h for NM and 48 h for Tau, cells were fixed with 4% paraformaldehyde, and nuclei were counterstained with Hoechst. Cells were imaged with the automated confocal microscope CellVoyager CV6000 (Yokogawa Inc.) using a ×20 or ×40 objective. Maximum intensity projections were generated from Z-stacks. Images from 16 fields per well were taken. At least $3–4 \times 10^3$ cells per well and at least 3 wells per treatment were analyzed.

For the coculture of HEK Tau-GFP[AD] cells with astrocytes, human primary astrocytes (ScienCell, # 1800) were transduced with lentivirus coding for Tau-FR after 24 h cultivation on 96-well plates. After 6 days, HEK Tau-GFP[AD] cells transiently expressing HERV-K, Syncytin-1, or mock-transfected were seeded onto astrocytes (donor: recipient, 5:1). The average number of cells per well was $2.8 \times 10^4$. Cells were cocultured for 24 h and subsequently fixed with 4% paraformaldehyde. Nuclei were counterstained with Hoechst. Cells were imaged using a ×40 objective. Maximum intensity projections were generated from Z-stacks. Images from 16 fields per well were taken. At least 900 cells per well and at least 3 wells per treatment were analyzed using ImageJ.

## Production of recombinant NM

To purify recombinant NM-His, BL21 (DE3) competent *E. coli* were transformed with 100 ng pET vector containing the coding sequence of NM with a C-terminal His-tag under the control of the T7 promoter. Five ml of *E. coli* overnight cultures were inoculated into 250 ml LB media containing 100 μg/ ml ampicillin. Cultures were incubated at 37 °C, 180 rpm (Multitron, Infors HT), until reaching an $OD_{600}$ of 0.8. NM-His expression was induced with 1 mM IPTG for 3 h at 37 °C, 180 rpm. Pellets (10 min, $3000 \times g$) from 1.5 l bacterial culture were pooled and lysed in 75 ml buffer A (8 M urea, 20 mM imidazole in phosphate buffer) for 1 h at RT. After sonication for $3 \times 10$ s at 50% intensity, cell debris was pelleted for 20 min at $10,000 \times g$ and the supernatant was sterile-filtered. NM-His was purified from the supernatant via IMAC using the ÄKTA pure protein purification system (GE Healthcare) together with a 5 ml HisTrap HP His tag protein purification column (GE Healthcare). The supernatant was loaded onto the column and initially washed with 25 ml buffer A. After rinsing with 75 ml buffer A, NM-His was eluted using a linear imidazole gradient from 10 to 250 mM imidazole (2–50% buffer B; 8 M urea, 500 mM imidazole in phosphate buffer). NM-His containing fractions were pooled and concentrated to around 10% of the initial volume using Vivaspin 20 concentrator columns (Sartorius, Germany) with a molecular cut-off of 10,000 Da. The protein was desalted using a 5 ml HiTrap Desalting column (GE Healthcare) and sterile-filtered PBS. Protein-containing fractions were pooled and frozen at −80 °C.

## Determination of size and number of EVs

ZetaView PMX 110-SZ-488 nano particle tracking analyzer (Particle Metrix GmbH) was used to determine the size and number of isolated EVs. The instrument captures the movement of extracellular particles by using a laser scattering microscope combined with a video camera. For each measurement, the video data is calculated by the instrument and results in a velocity and size distribution of the particles. For nanoparticle tracking analysis, the Brownian motion of the vesicles from each sample was followed at 22 °C with properly adjusted equal shutter and gain. At least three individual measurements of 11 positions within the measurement cell and around 2200 traced particles in each measurement were detected for each sample.

## Sample preparation for mass spectrometry

Five replicates of cell pellets and six replicates of EV samples from N2a NM-HA[agg] subclone s2E at passages 7 and 16 were collected for quantitative proteomics analysis. Cell pellets were lysed in 150 μL SDT buffer (4% SDS (w/v), 100 mM Tris/HCl pH 7.6, 0.1 M DTT) by homogenization with a Dounce tissue grinder and heated for 3 min at 95 °C. Afterward, the samples were sonicated 5 times for 30 s with intermediate cooling using a vialtweeter sonifier (amplitude 100%, duty cycle 50%; Hielscher, Germany). EV pellets were lysed in 100 μL STET lysis buffer

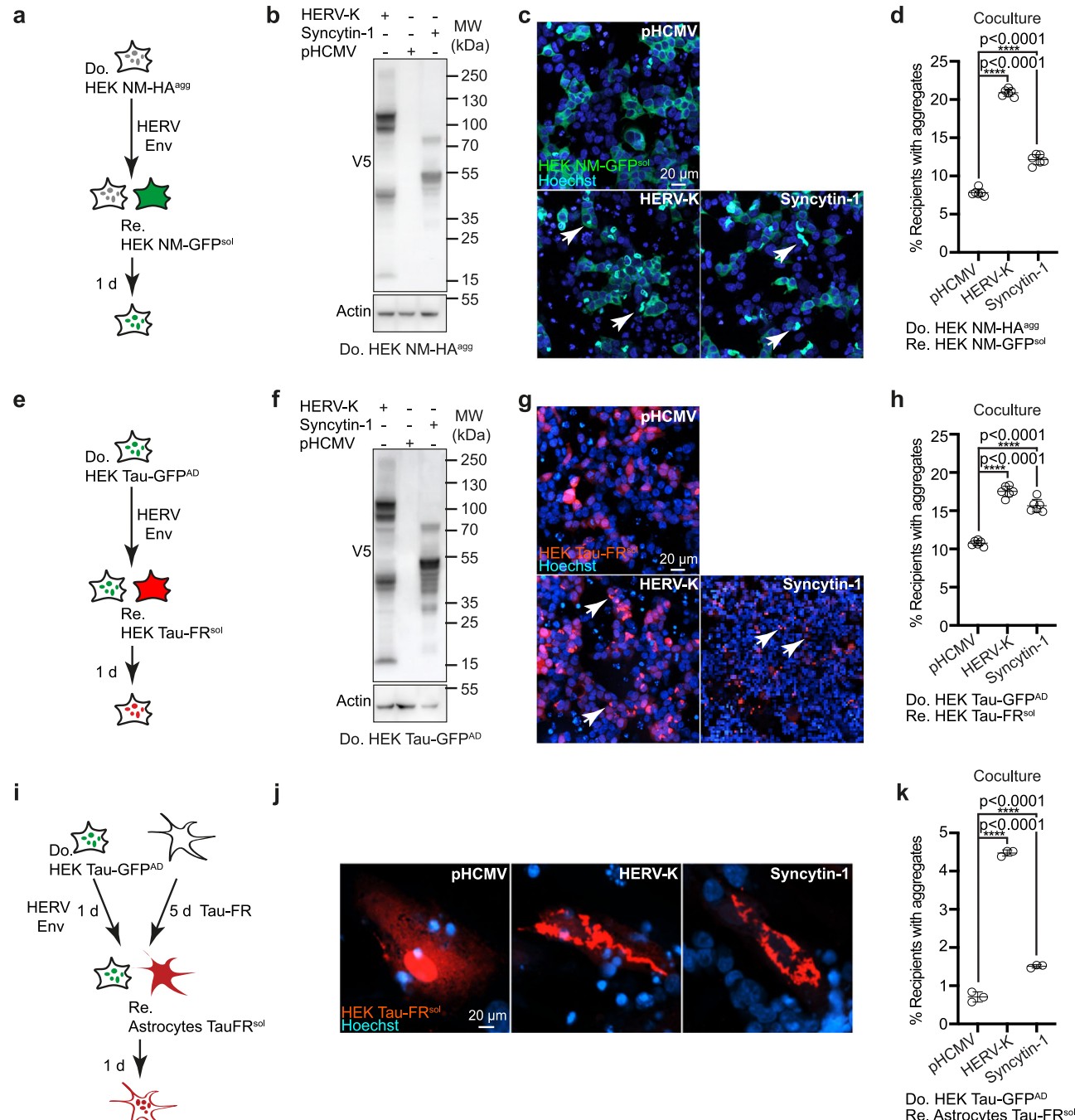

**Fig. 7 | Different HERV glycoproteins increase intercellular protein aggregate spreading. a** Experimental workflow. Donor HEK cells stably propagating aggregated NM-HA were transfected with a plasmid coding for V5-epitope tagged HERV-W Syncytin-1 or a plasmid coding for V5-epitope tagged HERV-K Env. Cells were subsequently cocultured with recipient HEK NM-GFP$^{sol}$. **b** Western blot analysis of donor clone transfected with plasmids coding for V5-tagged Syncytin-1 or HERV-K Env. Samples were loaded twice for Actin detection on different blots. **c** Coculture of donor and recipient HEK cells. Note that we have not stained the donors in this experiment. **d** Quantitative analysis of the percentage of recipient cells with induced aggregates upon coculture. **e** Experimental workflow. Tau-GFP$^{AD}$ cells were transfected with plasmids coding for V5 epitope-tagged HERV Envs and cells were subsequently cocultured with recipient HEK cells expressing Tau-FR$^{sol}$. **f** Western blot analysis of donor clone transfected with plasmids coding for HERV Envs.

**g** Coculture of donor and recipient cells. **h** Quantitative analysis of the percentage of recipient cells with induced aggregates. **i** Donor HEK Tau-GFP$^{AD}$ cells transfected or not with plasmids coding for HERV-K Env and -HERV-W Syncytin-1 were subsequently cocultured with human primary astrocytes expressing Tau-FR$^{sol}$. Please note that due to technical challenges, the high percentages of HERV Env expressing donor astrocytes with Tau aggregates required for cocultures cannot be achieved. **j** Primary astrocytes with soluble or aggregated Tau-FR following coculture. **k** Quantitative analysis of primary astrocytes with Tau-FR$^{agg}$ following coculture with HEK Tau-GFP$^{AD}$. All data are shown as the means ± SD from six (**d**, **h**) or three (**k**) replicate cell cultures. Three (**d**, **h**) and two (**k**) independent experiments were carried out with similar results. *P*-values were calculated by one-way ANOVA with Dunnett´s post hoc test (**d**, **h**, **k**). Source data are provided as a Source Data file.

(150 mM NaCl, 50 mM Tris–HCl pH 7.5, 2 mM EDTA, 1% Triton X-100) on ice for 30 min with intermediate vortexing. Cell debris and undissolved material were removed at $16,000 \times g$ for 5 min. The protein concentrations were measured using the colorimetric 660 nm assay (Thermo Fisher Scientific, USA). For cell lysates, the assay solution was supplemented with the ionic detergent compatibility reagent (Thermo Fisher Scientific, USA). A protein amount of 30 μg per sample for cell lysates and 10 μg for EV lysates was subjected to proteolytic digestion using the filter-aided sample preparation (FASP) protocol[75] with 30 kDa Vivacon spin filters (Sartorius, Germany). Proteolytic peptides were desalted by stop-and-go extraction (STAGE) with C18 tips[76]. The purified peptides were dried by vacuum centrifugation. Digestions of cell lysates and EVs were dissolved in 40 or 20 μl of 0.1% formic acid, respectively.

## LC–MS/MS analysis
Samples were analyzed by LC–MS/MS for relative label-free protein quantification. A peptide amount of approximately 1 μg per sample was separated on a nanoLC system (EASY-nLC 1000, Proxeon−part of Thermo Fisher Scientific, USA) using in-house packed C18 columns (50 cm or 30 cm × 75 μm ID, ReproSil-Pur 120 C18-AQ, 1.9 μm, Dr. Maisch GmbH, Germany) with a binary gradient of water (A) and acetonitrile (B) containing 0.1% formic acid at 50 °C column temperature and a flow rate of 250 nl/min. Cell lysates were separated on a 50 cm column using a gradient of 250 min length, whereas a 183 min gradient on a 30 cm column was used for EV samples (250 min gradient: 0 min, 2% B; 5 min, 5% B; 185 min, 25% B; 230 min, 35% B; 250 min, 60% B; 183 min gradient: 0 min, 2% B; 3:30 min, 5% B; 137:30 min, 25% B; 168:30 min, 35% B; 182:30 min, 60% B). The nanoLC was coupled online via a nanospray flex ion source (Proxeon−part of Thermo Fisher Scientific, USA) equipped with a PRSO-V2 column oven (Sonation, Germany) to a Q-Exactive mass spectrometer (Thermo Fisher Scientific, USA). Full MS spectra were acquired at a resolution of 70,000. The top 10 peptide ions were chosen for Higher-energy C-trap Dissociation (HCD) with a normalized collision energy of 25%. Fragment ion spectra were acquired at a resolution of 17,500. A dynamic exclusion of 120 s was used for peptide fragmentation.

## Data analysis and label free quantification
The raw data was analyzed by the software Maxquant (maxquant.org, Max-Planck Institute Munich) version and 1.5.5.1[77]. The MS data were searched against a FASTA database of *Mus musculus* from UniProt including also non-reviewed entries supplemented with databases of lentiviruses and MLVs (download: December 9, 2017, 52041 + 712 + 43 entries). Trypsin was defined as protease. Two missed cleavages were allowed for the database search. The option first search was used to recalibrate the peptide masses within a window of 20 ppm. For the main search peptide and peptide fragment mass tolerances were set to 4.5 and 20 ppm, respectively. Carbamidomethylation of cysteine was defined as a static modification. Acetylation of the protein N-term, as well as oxidation of methionine, were set as variable modifications. The false discovery rate for both peptides and proteins was adjusted to <1%. Label-free quantification (LFQ) of proteins required at least two ratio counts of razor peptides. Only unique and razor peptides were used for quantification. The LFQ values were $\log_2$- transformed and a two-sided Student's $t$-test was used to evaluate the statistically significant changed abundance of proteins between cell lysates from passages 16 and 7 as well as EV lysates from passages 15 and 6. A $P$-value <5% was set as the significance threshold. Additionally, a permutation-based false discovery rate estimation was used to account for multiple hypotheses[78].

## OptiPrep density gradient
For separating EVs and virus, a discontinuous iodixanol gradient in 1.2% increments ranging from 6% to 18% was used. The $100,000 \times g$ pellet from 1050 ml culture supernatant (30 × T175 flasks) was resuspended in 1 ml PBS and overlaid onto the gradient. The gradient was subjected to high-speed centrifugation at $100,000 \times g$ for 2 h at 4 °C using a SW41Ti rotor (Beckman Coulter). 12 fractions of 1 ml each were collected from the top of the gradient, diluted with PBS in 5 ml and centrifuged at $100,000 \times g$ for 1 h at 4 °C. The pelleted fractions were resuspended in 100 μl PBS and then used for further experiments. The reverse transcriptase activity of the viruses was measured by using Reverse Transcriptase Assay, colorimetric (Roche).

## Electron microscopy (EM)
For EM imaging of EVs and virus preparations, the $100,000 \times g$ pellets were fixed in 2% paraformaldehyde, loaded on glow-discharged Formvar/carbon-coated EM grids (Plano GmbH), contrasted in uranyloxalat (pH 7) for 5 min and embedded in uranyl-methylcellulose for 5 min. Samples were examined using a JEOL JEM-2200FS transmission electron microscope at 200 kV (JEOL).

## Infectivity assay
For the infectivity assay, Melan-a cells were exposed to conditioned medium from different cell clones at either low or high passages in the presence of 4 μg polybrene/ml for 24 h. The medium was then replaced with a normal culture medium. After 5 days, cells were lysed for Western blot analysis for retroviral Env and Gag proteins.

## Drug treatment
The treatment of cells with Amprenavir (10 μM; Selleckchem), Lopinavir (10 μM; Selleckchem), and DMSO was performed for 72 h in EV-depleted medium in T175 flasks. Afterward, the total number and the viability upon drug treatments were determined using the Vi-VELL™XR Cell Viability Analyzer (Beckman Coulter). EVs were isolated from the conditioned medium via ultracentrifugation and processed for the aggregate induction assay as described above. Coculture and EV aggregate induction for NM were performed in the absence of the drugs, whereas all cell-based assays were performed in the presence of drugs at the same concentration as the pretreatment. To inhibit methyltransferase, N2a NM-HA$^{agg}$ P1 donor cells were treated for 3 days with methyltransferase inhibitors 5-Azacytidine (Aza) (Sigma-Aldrich), Decitabine (Dec) (Sigma-Aldrich) or DMSO as solvent control. Subsequently, the cells were cultured in the absence of the drugs for 5 days. Afterward, the treated donor cells were cocultured with recipient cells as described above to assess their aggregate-inducing capacity and analyzed by Western blot for MLV Env and Gag expression. To increase DNA methylation, the N2a NM-HA$^{agg}$ P21 donor clone was treated with methyl group donors L-methionine (L-M, Selleckchem), betaine (B, Selleckchem), choline chloride (CC, Selleckchem) or medium control for 6 days. MLV Env and Gag protein levels were analyzed by Western blot. Subsequently, cells were cocultured with recipient cells for 16 h. The percentage of aggregate-containing recipient cells was compared to the percentage of aggregate-bearing recipients cocultured with solvent-treated donors.

## Neutralization assay
To block MLV Env on the surface of the donor cell clone and EVs, mAb83A25, which reacts with almost all members of MLVs[79] was incubated with either EVs or donor cells in serial dilutions for 1 h at 37 °C with rotation at 20 rpm. Afterward, the donor cells were mixed with recipient cells and EVs were added to the pre-seeded recipient cells for 1 day.

## Transfection with siRNAs or plasmids
To transiently knock down the upregulated specific MLV Env and Gag genes in N2a NM-HA$^{agg}$ donors, custom-designed Silencer select siRNAs from Thermo Fisher against AAO37244.2 (*env*) and AID54952 (*gag*) were used. Pre-designed siRNAs from Thermo Fisher Scientific against

murine XPR1 and mCat-1 genes were used to knock down both genes. For transfection, cells were pre-seeded on a six-well plate 1 day before at $2 \times 10^5$ cells/well. The next day, cells were transfected with a final concentration of 60 nM siRNA or control siRNA using lipofectamine RNAiMax diluted in Opti-MEM for 30 min before addition to cells. After 2–3 days, cells were collected for aggregate induction assays, qRT-PCR, or Western blot analysis. siRNAs were designed to target X/P-MLV *env*, siRNAs against MLV *gag* were designed to target X/P- and E-MLV.

## qRT-PCR

Validation of changes of different transcripts in cell pellets was carried out by real-time quantitative polymerase chain reaction (qPCR). First, total RNAs from cell pellets were isolated using the RNeasy Mini Kit or RNeasy Lipid Tissue Mini Kit (Qiagen). RNA concentration and quality were determined with Agilent 2100 Bioanalyzer System. RNAs were reversely transcribed to cDNA using the iScript™ cDNA Synthesis Kit (Bio-Rad). To determine mRNA expression levels of murine Env AAO37244.2 and Gag AID54952, custom-designed TaqMan assays were used. To identify polytropic *env* transcripts, Taqman assays were based on the provided genomic sequence AY219544.2 (nucleotides 2786-3298). This sequence codes for an Env which shares 100% amino acid similarity with our polytropic Env (P10404: 100% similarity to AAO37244.2 Env protein encoded by the AY219544.2 nucleotide sequence). An NCBI nucleotide blast search using the respective *env* nucleotide region reveals 100 hits with sequence similarity of at least 99.6%. Infectious P-MLV viruses carry ecotropic MLV backbones with P-MLV-related changes in the 3′ *pol* and 5′ *env* regions[48]. Genomic sequence J01998.1 coding for ecotropic MLV, nucleotides 756–1219, was provided for Taqman MLV *gag* assay design. This region shares at least 93% similarity with ~90 ecotropic, xenotropic, and polytropic MLV *env* sequences according to NCBI nucleotide blast. For murine *xpr1*, *mcat-1*, and *gapdh* as housekeeping control pre-designed TaqMan assays were used. For detection of the MLV subclasses, mRNA expression levels were measured using SYBR Green assays (Applied Biosystems). For quantitative real-time PCR, the fold change was calculated with the ΔΔCT method. The following primers were used: *X/P-tropic* MLV (forward: 5′-GGAGCCTACCAAGCACTCA-3′; reverse: 5′-GGCAGAGGTATGGTTGGAGTAG-3′). These primers bind with a maximum of two mismatches to X/P-MLVs[20]. *Ecotropic* MLV (forward:5′-AGGCTGTTCCAGAGATTGTG−3′; reverse: 5′-CCGGGGCAGACATA-GAATCC-3′)[80]. These primers do not bind to X/P-MLVs[20]. qPCR primers for *ETnI, ETnII, MusD, IAP*, and *LINE1 ORFp* were according to[81]. qPCR primer pairs for murine cytokines *TNF-α, IRF3*, and *IL6* were from ref. [82]. pPCR primers for murine *IFN-ß* were from Origene (#MP206681). Primer pairs for human cytokines were from Origene: *IFN-ß* (#HP205913); *IL1ß* (#HP200544); *TNF-α* (#HP200561); *IRF3* (#HP205426); *IL6* (#HP200567). GAPDH (forward: 5′-ACCCAGAA-GACTGTGGATGG-3′; reverse: 5′-TCAGCTCAGGGATGACCTTG-3′).

*GAPDH* (forward: 5′-GCACAGTCAAGGCCGAGAAT-3′; reverse: 5′-GCCTTCTCCATGGTGGTGAA-3′).

## Western blotting

For Western blot analysis, protein concentrations were measured by Quick Start™ Bradford Protein assay (Bio-Rad) using the plate reader Fluostar Omega BMG (BMG Labtech) and the corresponding MARS Data Analysis Software (BMG Labtech). Proteins were separated on NuPAGE®Novex® 4-12% Bis−Tris Protein Gels (Life Technologies) followed by transfer onto a PVDF membrane (GE Healthcare) in a wet blotting chamber (BioRad). Western blot analysis was performed using rat hybridoma anti-MLV Env mAb83A25[79] (1:10; kindly provided by L.H. Evans, Rocky Mountain Laboratories, MT); goat anti-xenotropic MLV virus antibody ABIN457298 for detecting both Env and Gag (1:1000, antibodies-online); mouse anti-MLV Gag ab100970 (1:1000, Abcam); rat anti-HA 3F10 (1:1000, Roche); mouse anti-GAPDH 6C5 (1:5000, Abcam); mouse anti-Actin C4 (1:5000; MP Biomedical); rabbit anti-Tau

ab64193 (1:1000, Abcam); mouse anti-PiT-2 B-4 (1:1000; Santa Cruz Biotechnology); rabbit anti-V5 D3H8Q (1:1000, Cell signaling); mouse anti-Alix (1:1000, BD Bioscience); mouse anti-Hsc/Hsp70 N27F3-4 (1:1000, ENZO); rabbit anti-Flotillin1 ab133497 (1:1000, Abcam). The membrane was incubated with Pierce™ ECL Western Blotting Substrate (Thermo Fisher Scientific) according to the manufacturer´s recommendations and imaged with the Imaging system Fusion FX (Vilber Lourmat). Unprocessed and uncropped scans are provided in the Source Data file.

## Automated image analysis

The image analysis was performed using the CellVoyager Analysis support software (CV7000 Analysis Software; Version 3.5.1.18). An image analysis routine was developed for single-cell segmentation and aggregate identification (Yokogawa Inc.) The total number of cells was determined based on the Hoechst signal, and recipient cells were detected by their GFP/FR signal. Respective green or red aggregates were identified via morphology and intensity characteristics. The percentage of recipient cells with aggregated NM-GFP or Tau-FR was calculated as the number of aggregate-positive cells per total recipient cells set to 100%.

## Immunofluorescence staining and confocal microscopy analysis

Cells were fixed in 4% paraformaldehyde. For HA-staining cells were rinsed with PBS, permeabilized in 0.1% Triton X-100, blocked in 2% goat serum in PBS, and incubated with rabbit anti-HA 3F10 (Roche) antibody diluted 1:500 in blocking solution for 2 h at RT. After three washing steps with PBS, cells were incubated for 1 h with Alexa Fluor 647-conjugated secondary antibody A21247 (1:800, Invitrogen), while nuclei were counterstained with 4 µg/ml Hoechst 33342 (Molecular Probes). 384-well plates were scanned with CellVoyager CV6000 (Yokogawa Inc.). Confocal laser scanning microscopy was performed on a Zeiss LSM 800 laser-scanning microscope with Airyscan (Carl Zeiss) and analyzed via Zen2010 (ZenBlue, Zeiss).

## Statistical analysis

All analyses were performed using Prism 6.0 (GraphPad Software v.7.0c). For multiple comparisons of more than two groups, we performed one-way ANOVA with Tukey's multiple comparisons tests. When comparing selected pairs of means, with the selection based on experimental design, we have used Bonferroni's post hoc test. For pairwise comparisons of multiple treatment groups with a single control group, we have used one-way ANOVA with Dunnett's post hoc test. For the comparison of the two groups, we used a two-sided Student's *t*-test. The confidence interval in both tests was 95% and P-value < 0.05 were considered significant. All experiments were performed in at least triplicates (EV experiments) or at least sextuplicates (coculture experiments) and repeated at least two times independently with similar results. Measurements were taken from distinct samples. At least 6000 cells were analyzed for quantitative analysis. Shown are the mean and the error bar representing the standard deviation (SD).

## Reporting summary

Further information on research design is available in the Nature Portfolio Reporting Summary linked to this article.

# Data availability

The authors declare that all data supporting the findings described in this paper are available within the article and its supplementary information files. The mass spectrometry proteomics data have been deposited in the ProteomeXchange Consortium via the PRIDE[83] partner repository with the dataset identifier PXD043201. Source data are provided with this paper.

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

## Acknowledgements

We thank Pug Evans for generously sharing anti-MLV antibodies. We are grateful to Paolo Salomoni and Dan Ehninger for the critical reading of the manuscript. The light microscopy (LMF) and laboratory automation facilities (LAT) of the DZNE Bonn were used for image acquisition. This work was funded by the Helmholtz Portfolio "Wirkstoffforschung" (I.M.V.), the "Deutsche Forschungsgemeinschaft" (DFG, German Research Foundation) under Germany's Excellence Strategy within the framework of the "Munich Cluster for Systems Neurology" (EXC 2145 SyNergy–ID 390857198) (S.L.), by the German Ministry for Education and research through grants CLINSPECT-M (FKZ161L0214C) (S.L.) and JPND PMG-AD (01ED2002B) (S.L.). The funders had no role in study design, data collection, and analysis, the decision to publish, or the preparation of the manuscript. Open Access funding enabled and organized by project DEAL.

## Author contributions

S.L., S.-E.H., S.F.L., and I.M.V. designed research; S.L., S.-E.H., A.H., O.B., S.A.M. performed research; S.F.L., S.A.M., P.D., and I.M.V. analyzed data; S.L., S.-E.H., and I.M.V. wrote the paper.

## Funding

## Competing interests

S.L., S.A.M., S.F.L., P.D., and I.M.V. hold pending patent applications for "HERV inhibitors for use in treating tauopathies": "US Patent Application No. 17/640,119 based on PCT International Application No. PCT/EP2020/ 074809, claiming priority to "European Application No. 19195304.1". The remaining authors declare no competing interests.
