## [Peer Review File · Nature Communications]

Reactivated endogenous retroviruses promote protein aggregate spreadingREVIEWER COMMENTS

Reviewer #1 (Remarks to the Author):

The study by Liu and colleagues provides experimental evidence on a novel role of endogenous retroviruses (ERVs) in contributing to pathological protein folding and to so-called seed spreading. The background consists in the observation that misfolded proteins such as Tau, TDP43 or FUS accumulate as insoluble aggregates in patients' brains and were shown/supposed to contribute to pathologies such as AD, ALS or frontotemporal lobar degeneration. Such misfolded protein aggregates can serve as templates for yet correctly folded monomers similarly to what has been described for prions. The authors have now studied in culture cell-to-cell spreading of such protein aggregates in the context of (co-expressed) endogenous retroviruses or their encoded proteins.

Using a number of experiments mainly using mouse neuroblastoma cells, they demonstrated that i) ERV upregulation in donor cells fosters intracellular aggregate production (of yeast Sup 35 domain NM) in recipient cells, ii) mouse retrovirus MLV proteins to be induced in donor cells as well as in their extracellular vesicles, iii) rescue by the anti-MLV Env antibody, iv) modulation via RNA interference, epigenetics and retroviral protease inhibitors and v) involvement of the MLR receptor XPR1. In addition, using human embryonic kidney (HEK) and Vero (monkey kidney) cell lines they revealed a similar transmission and induction of Tau protein aggregates and finally two human endogenous retrovirus (HERVs), ERVW1/Syncytin-1 and HERV-K HML-2, i.e., their envelopes were found to act similarly on Sup 35 domain NM and Tau protein aggregation (again in HEK and Vero cells).

This is a really interesting study which could contribute to the emerging recognition of human endogenous retrovirus' (HERV) relevance in different neurological and psychiatric pathologies.

The major limitation of the study is related to the rather artificial experimental setup including mainly cells lines (N2a, HEK, Vero) for most of the experiments in combination with detailed investigations on mostly a yeast derived aggregating protein Sup35 as well as the study of not necessarily pathogenic HERV variants. Moreover, a single N2a subclone s2E was selected for improved aggregate induction, which is also critical. Only for data shown in Fig.1 primary mouse cortical neurons were used as additional recipients. This setup has to be challenged in order to provide human disease relevant data. In this regard it would also be highly supportive if aggregate formation could be induced in donor cells using AD brain homogenates. Can HERV proteins also be found in such homogenates?

Being aware of the current lack or shortage of suitable in vivo HERV models and given that particularly the pathogenic HERV entities are human-specific, I would nevertheless prefer if corresponding experiments could be performed using as close as possible parameters relevant for the human diseases. This encompasses human primary or iPSC derived neuronal cells (as donors as well as recipients) and not cell lines. In this context I wonder whether only a neuron-to-neuron transfer is relevant and whether glial cells as donors or recipients are also conceivable.

Moreover, HERV-K variant HML-2 and HERV-W variant pV14 revealed to be of relevance for a number of human pathologies. So, the MSR-V-pV14 (GenBank, AF331500) encoding the corresponding HERV-W variant should be considered and tested. Syncitins are rather inappropriate as these are domesticated proteins with physiological functions and to my knowledge no CNS expression was shown so far. Please also consider that Charvet et al., 2021 showed that the pathogenic form of HERV-W present in multiple sclerosis (AF331500) leads to different oligomerization compared to physiological Syncytin-1.

Note, HML-2 Env was recently found to bind CD98HC complexed to β 1 integrin (Steiner et al., 2022), pHERV-W (pV14) was shown to bind TLR4 (Madeira et al., 2016). This must also be considered in the study of appropriate recipient cells.

There are many more mouse endogenous retroviral elements known such as ETnI, ETnII- α , ETnII- β , ETnII- γ , MusD and IAP (Cipriani et al., 2018), often not listed in databases. Could the authors comment on this fact? Were you able to detect these retroviral elements in your proteome

analysis? Were they regulated?

HERVs have multiple times been shown to induce inflammation and immunological responses. Have the authors analysed any inflammatory or immunological markers in this context? Even though it is not the leading hypothesis of protein aggregation in AD and PD, there is certain literature (e.g. Sanchez de Groot and Torrent Burgas, 2015) hypothesizing that inflammation might lead to miss-folding of proteins and therefore increasing protein aggregation. Can inflammation as cause for the protein aggregation be excluded?

Regarding your statistics, none of your datasets were tested for normal distribution. However, all of your statistical test assume that your datasets are normally distributed. Although mostly showing low numbers of replicates, the Shapiro-Wilk test should be able to test your datasets for normality. If you have performed a test for normal distribution, it should be described in your material and method section, if not please revise your manuscript accordingly. Furthermore, I do not really understand why you are switching from one-way ANOVA with Bonferroni post-test to Students t-test back and forth. The manuscript would benefit from a more clear-cut way of performing statistics (and actually multiple students t-testing is not the best way). GraphPad 6 might not be able to do that, but Bonferroni is also a rather weak statistical post hoc test. Sidak or Tukey would be more powerful.

Could the authors comment on the fact that they often switch between normalized and non-normalized data regarding the "% of recipients". I think it is rather confusing the reader.

Minor comments:

Legend Fig. 1c is inconsistent with graph labelling: "c. Percentage of recipient cells with induced NM-GFP aggregates upon coculture with donor N2a NM-HAagg cells P6 and P17."

Fig. 3b: awkward presentation of lowered levels upon siRNA ("fold change over control") use instead bars also for control levels so that effects can intuitively been recognized. This is also true for most of the data shown in Fig. 3.

Fig. 3d: Why are there two graphs labeled with cocultures shown? Would the right one not be EVs?

Legend Fig. 3: "were transfected"

Fig. 4: The figure legend of Fig. 4 might still contain some writing artefacts "(f????)"?

Fig. 5: Awkward demonstration of receptor downregulation in Fig. 5b., which is also true for 5c and 5d; avoid "normalized to control".

Fig. 6: Figures 6c-e should best be combined in one graph, Figures 6i and m and n should also be combined.

Reviewer #2 (Remarks to the Author):

Studies like Liu et al. deserve full attention and support. Why? Over recent decades, there has been great progress in understanding the molecular details of familial forms of neurodegeneration and dementia. Yet, spontaneous, not inherited, cases represent the epidemiological bulk. Their mechanisms are only rudimentary known when compared to our understanding of the familial neurodegeneration. The burgeoning field of neuroinflammation has recently begun to correct this imbalance. Previous work of Vorberg's group indicated that viruses, actually infectious agents, can support the spread of aggregates even without induction of inflammation. In the current study, they continue the line by discovering that and explaining how the reactivated endogenous retroviruses can do this as well. To facilitate the proper perception of the importance of these findings by general readership I would recommend to simplify the title a little bit, for example

“Reactivated endogenous retroviruses promote aggregate spreading” or similar.

While reading the manuscript, I was looking for experimental evidence in support of some kind of signaling-related mechanism(s) and could not find any. On the contrary, I became convinced of the structural role of reactivated retroviruses as drivers and vehicles that mediate the spread of seeds of aggregation in the cell culture models used by the authors. The clarification of the data presented Fig.2h might be helpful in consolidating this impression. Specifically, how does the cumulative aggregate-spreading “potency” of the fractions 2 to 6 compare to the original potency of the conditioned medium used for pelleting and fractionation? If fractions can do the job only weakly, is there a possibility of additional (soluble) factors that are lost during pelleting and/or fractionation? Regardless of these optional considerations, I found the fractionation experiments very important in view of future work. It is quite obvious that structural biologists will be keen to analyze the structural characteristics of proteopathic seeds transported in the donor cell-generated extracellular vesicles in this experimental model.

What I could not understand is the fact that only a certain fraction, typically 30-40%, of recipient cells converts their soluble aggregation-prone proteins to real aggregates. Does the vulnerability depend on the cell cycle status of the exposed recipient cells? What are other possible explanations? Could the authors discuss these possibilities in the manuscript? Why the XPR1-transfected HEK293T cells reach the similar efficiency as N2a cells (Fig. 5h)? Does it mean that the transfection efficiency reaches 100%? How efficient would be the conversion of recipient cells with a lower transfection rate? Although the conversion does not happen in all cells, I do strongly support the simplified rendering of the experimental setups as provided by the authors in almost all figures. These schemes make following the text and data easy, despite the complexity of the performed experiments. Also more generally, the flow of the text and the argumentation are well structured and clear and a pleasure to read (except of the last paragraph of the Discussion).

Several minor issues:

Fig. 1h vs. 1j: mRNAs of gag and, especially, env in cells are upregulated much more than the respective proteins. Is there a problem with translation of the viral mRNAs in the system?

Fig. 1g: please indicate the gag processing products MA and NC as they are mentioned in the text (lane 160).

Fig. 4c: swap the upper and lower schemes to make the alignment with the respective curves in Fig. 4d more natural.

Fig. 4 legend, lane 1254: (f????) needs to be corrected.

Fig. 5b: delete XPR1/mCat-1 from the Y-axis label, otherwise confusing.

Fig. 5e: provide continuous sequences of the loops.

Fig. 6c-e: set the same Y-axis scale for the ease of comparison.

Fig. 6l-m: as above.

I strongly support publication of this study.

Reviewer #3 (Remarks to the Author):

In this manuscript, the authors have explored that up-regulation of the murine endogenous retroviral (MLV) proteins env and gag results in intracellular aggregate induction as well as aggregate spreading. Targeting MLV maturation using protease inhibitors impaired intracellular protein aggregation. Moreover, overexpression of specific HERV glycoproteins increased intercellular protein aggregation. However, the study unfortunately lacks both the mechanism of how MLV gag/pol and env get increased as well as the mechanism of how MLV gag/pol and env lead to the induction of intracellular aggregates. Thus, the data and experiments presented in this study require substantial additional changes before publication.

Major Comments:

1) Since only gag and env of MLV were found upregulated in the proteomic analyses and these were then also examined throughout the manuscript, the current manuscript title is misleading as

it implicates that all ERVs affect intracellular aggregates. This should be made more specific.

2) To show that only MLV gag/pol and env are affecting intracellular NM aggregation and aggregate spreading it would be necessary to also investigate the cells on transcript levels (RNAseq) beside the proteomics approach that was conducted. This would show if other transcripts not coding for env or gag would also play roles in the observed findings. This is not unlikely, since the vast majority of HERV elements in our genome consist of solitary LTRs and mutant HERV proviruses. In addition, it could also help to provide a better insight into the mechanisms that affect intracellular NM aggregation and aggregate spreading.

3) The authors show that there is an increase in MLV gag and env proteins in specific neuroblastoma Sup35 NM mouse model. However, several things are unclear concerning the experiments in which MLV gag or env are downregulated by siRNAs or reactivated by epigenetic drugs to show that they are related to intracellular NM aggregation:

a) After knockdown with the specific MLV siRNAs, RNA (Fig 3B) and protein (Fig3 C) levels behave very differently. While on the RNA level only a 50% knockdown with an MLV env siRNA can be detected, on the protein level a much more efficient knockdown can be detected. For MLV gag it behaves the opposite. Here a more efficient knockdown is detectable on RNA level whereas on Protein level nearly no knockdown is visible (considering that also GAPDH is affected here). How can this be explained? In addition, the WB's (Fig 3C, G, J) must be quantified so that a comparison is easy and changes in GAPDH levels are considered.

b) The authors justify the weak knockdown via existing multicopy ERVs. Which MLVs copies are exactly targeted by the siRNAs? The qPCR Primers are targeting which copies?

c) How do the authors exclude off target effects by using siRNAs against multiple copies of MLV gag or env genes? It would be necessary to at least show that MLV gag protein levels are unaffected upon siRNA treatment against MLV env and vice versa for the siRNAs against MLV gag.

d) The WB for gag in Fig 3C, G and J is cut. Therefore, it is not possible to see the different gag bands. How do they behave? Are they also regulated by the siRNAs and epigenetic drugs? This would be interesting since they are strongly induced in Fig 1K.

4) The authors can show that upregulated MLVs affecting NM aggregate induction belong to the X/P-MLVs. These data suggest that these MLVs are the ones driving the observed effects. Therefore it would be interesting to target only X/P-MLVs specifically using siRNAs or other approaches (e.g. CRISPRi).

5) The authors show that expression of HERV-W env Syn-1 as well as HERV-K env K113 can increase the intracellular protein aggregation and spreading. To show that these effects are really HERV-W and HERV-K env specific the authors should also include an HERV derived envelope from a HERV group, which has not been associated with neuropathy.

Minor Comments:

1) Supplementary Table 1 was unfortunately not included or accessible.

2) Why does the env expression behave so different in the control cells of Figure 3G compared to the control cells of Figure 3C and J? The lower band here is much weaker, which then naturally entails a rise to Aza and Dec treatment. When comparing the increase to the other control cells, the increase in MLV Gag and Env would be more or less the same after Aza and Dec treatment. Again, quantification of the WBs is necessary for comparison. Beside, the increase in aggregates Fig 3H is significantly higher after Aza treatment than after Dec treatment. How can this be explained? Could this be due to other, non MLV Env/Gag effects?

3) In Fig 4B it would be easier to follow the different protease inhibitors if a color code would be used.

4) How do gag and env levels behave in Fig 5d and i?

5) Line 289-290: "Highest induction rates in cocultures were observed when donor cells were also transfected with retroviral transfer vector (TV) (Fig 6e)." The induction in Fig 6e compared to Fig 6d is not increased if one compares +gag/pol, +env (Fig6D) and +TV, +gag/pol, +env (Fig6E). In both samples values are approx. 22-24.

POINT-BY-POINT RESPONSE TO REVIEWERS

Liu et al. (# NCOMMS-22-24759A-Z):

“Endogenous retroviruses promote prion-like spreading of proteopathic seeds”,
title now changed to

“Reactivated endogenous retroviruses promote protein aggregate spreading“

We would like to thank the reviewers for their thoughtful and constructive input on our manuscript. We are very pleased with the reviewer comments and welcome the opportunity to submit a revised version of our work.

We have now performed a substantial number of supporting experiments that show the following:

- 1) Cytokines do not play a substantial role in intercellular aggregate spreading in our assays
- 2) Expression of ERV Env proteins does not induce spontaneous protein aggregation in our system
- 3) Envelope proteins of diverse HERV clades (K, W, R, H) increase the intercellular spreading of proteopathic seeds, including Tau
- 4) This process can be inhibited by blocking HERV receptors by competing viral ligands and can be enhanced by expression of HERV Envelope mutants with increased binding and fusogenic activity
- 5) HERV envelope proteins expressed by donors enhance protein aggregate spreading to primary human astrocytes

We have further performed additional control experiments (see below). All of these additional experiments support our conclusion that ERV gene products affect the intercellular spreading of proteopathic seeds. We have substantially revised the text as suggested by the reviewers and included a schematic on retroviral genome organisation, transcripts, protein maturation and virion composition, for clarity.

Please find enclosed a point-by-point response to the reviewer’s comments.

Reviewer #1 (Remarks to the Author):

The study by Liu and colleagues provides experimental evidence on a novel role of endogenous retroviruses (ERVs) in contributing to pathological protein folding and to so-called seed spreading. The background consists in the observation that misfolded proteins such as Tau, TDP43 or FUS accumulate as insoluble aggregates in patients’ brains and were shown/supposed to contribute to pathologies such as AD, ALS or frontotemporal lobar degeneration. Such misfolded protein aggregates can serve as templates for yet correctly folded monomers similarly to what has been described for prions. The authors have now studied in culture cell-to-cell spreading of such protein aggregates in the context of (co-expressed) endogenous retroviruses or their encoded proteins.

Using a number of experiments mainly using mouse neuroblastoma cells, they demonstrated that i) ERV upregulation in donor cells fosters intracellular aggregate production (of yeast Sup 35 domain NM) in recipient cells, ii) mouse retrovirus MLV proteins to be induced in donor cells as well as in their extracellular vesicles, iii) rescue by the anti-MLV Env antibody, iv) modulation via RNA interference, epigenetics and retroviral protease inhibitors

and v) involvement of the MLR receptor XPR1. In addition, using human embryonic kidney (HEK) and Vero (monkey kidney) cell lines they revealed a similar transmission and induction of Tau protein aggregates and finally two human endogenous retrovirus (HERVs), ERVW1/Syncytin-1 and HERV-K HML-2, i.e., their envelopes were found to act similarly on Sup 35 domain NM and Tau protein aggregation (again in HEK and Vero cells). This is a really interesting study which could contribute to the emerging recognition of human endogenous retrovirus' (HERV) relevance in different neurological and psychiatric pathologies.

Response: We thank the reviewer for his/her strong support of our study.

The major limitation of the study is related to the rather artificial experimental setup including mainly cells lines (N2a, HEK, Vero) for most of the experiments in combination with detailed investigations on mostly a yeast derived aggregating protein Sup35 as well as the study of not necessarily pathogenic HERV variants.

Response: A concern of the reviewer is that experiments were mostly performed with NM. We respectfully disagree. Our study is based on our curious finding that continuous culture of a specific donor cell clone with NM aggregates upregulates endogenous MLV. This was a purely circumstantial finding, which unfortunately cannot be replicated identically experimentally. Generally, the cellular events that trigger spontaneous ERV reactivation in cell culture (despite being known for decades) are unresolved¹⁻⁶.

However, we have confirmed in multiple experiments with different cell lines, expressing either NM or Tau, that viral gene products of the MLV family alone or in combination can increase intercellular protein aggregate dissemination. Thus, we provide evidence that MLV gene products increase the spreading also of disease-relevant protein Tau.

*We further demonstrated that expression of envelope proteins of HERV-W and HERV-K is sufficient to increase intercellular dissemination of both Tau and NM aggregates. We refrain from defining specific HERVs as non-pathogenic, as aberrant expression of diverse HERV clades and retroelements has been observed in disease. If and how HERV expression contributes to certain maladies remains to be established. We now include also experiments with envelope proteins of HERV clades H and R, again demonstrating increased Tau and NM aggregate induction in recipients (**new Supplementary Fig. 9 a-j**). Our experiments therefore argue that aberrant expression of diverse HERV Envs can mediate intercellular aggregate dissemination, a finding that has great relevance in light of the growing number of studies demonstrating reactivation of diverse HERV clades upon aging, inflammation and neurodegeneration.*

We agree with the reviewer that using differentiated human neurons or astrocytes or iPSC-derived cells would be highly desirable for our experiments. We would like to clarify why we are limited in cellular models to study intercellular protein dissemination: Experimental models to quantitatively analyze intercellular dissemination of protein aggregates are extremely challenging and cannot be easily adapted to primary cells. These require the generation of donor populations with high numbers of aggregate-bearing cells and donor and recipient cells that can tolerate multiple manipulations (transfections/ transductions/ fibril or brain homogenate exposures), medium exchange and cell splits. As an example, producing donors requires NM/ Tau expression, aggregate induction, Env expression and subsequent use in experiments. We usually produce such a donor population in which close to 100 % of cells contain aggregates by cloning cells with aggregates prior to using them in experiments. This procedure takes at least 2-4 months. With limited to poor transfection/transduction and aggregate induction efficiencies in primary cells, only a minor subpopulation of donors actually produces aggregates and expresses Env (if they survive the manipulations at all). Further, our attempts so far proved toxic to iPSCs. We are unaware of

primary cellular models with this degree of experimental manipulation that allow for quantitative analysis. We anticipate that adapting our quantitative assays to assays using primary cells/ iPSC-derived neurons or astrocytes as donors would require, if at all possible, months to years of development.

However, in addition to showing that mouse primary neurons can act as recipients, we now successfully demonstrate that human primary astrocytes expressing our Tau construct can produce aggregates when cocultured with HERV Env expressing donors (Fig. 7i-k).

Moreover, a single N2a subclone s2E was selected for improved aggregate induction, which is also critical.

Response: As pointed out above, reactivation of ERVs in cell culture is a poorly understood, spontaneous event¹⁻⁶. We found this cell clone by serendipity by testing close to 100 N2a NM-HA^{agg} cell clones for increased aggregate induction rates in cocultures. Careful standard operating procedures and proteomic analyses of cell lysates and EVs of this clone revealed that increased induction rates upon prolonged cell culture were likely due to reactivation of MLV, a hypothesis that we convincingly confirmed in subsequent functional assays. Importantly, reconstitution experiments in MLV-negative HEK cells demonstrate that MLV gene products alone or in combination increase intercellular spreading of NM and Tau aggregates.

Only for data shown in Fig.1 primary mouse cortical neurons were used as additional recipients. This setup has to be challenged in order to provide human disease relevant data. In this regard it would also be highly supportive if aggregate formation could be induced in donor cells using AD brain homogenates.

Response: Please note that in our MLV experiments, polymorphic variants of the human XPR1 receptor can inhibit MLV Env- receptor interactions (see Figure 5), so we included murine primary neurons for MLV (Fig. 1g). To provide human disease relevant data, we performed additional experiments demonstrating that our findings are relevant also for HERV gene products and recipient human primary astrocytes (Fig. 7i-k). As donor cells, we have used HEK Tau-GFP^{AD} cells. Tau-GFP aggregates in these cells had been induced using AD-patient brain homogenate⁷. Of note, initial induction rates prior to cloning of HEK cells for aggregate-bearing cells was below 5 % (not shown). Please note that due to the experimental challenges discussed above, we are unable to produce donor primary astrocytes with sufficient numbers of aggregate-bearing cells that sustain further required manipulations.

Can HERV proteins also be found in such homogenates?

Response: This is an interesting question. Independent studies have already demonstrated that HERV RNAs and proteins become upregulated in ND patients suffering from AD, Progressive Supranuclear Palsy, ALS, Behavioral Variant Frontotemporal Dementia, Creutzfeldt-Jakob disease and Multiple Sclerosis⁸⁻¹⁵. We have not demonstrated expression of HERVs in ND patients, as meaningful results can only be acquired by analyzing large numbers of patient samples, which we unfortunately currently do not have access to. Our focus was to test the hypothesis that ERV gene products contribute to protein aggregate spreading.

It has been reported that Tau aggregation can cause HERV expression¹²⁻¹⁴. We have now controlled for HERV Env protein in HEK Tau-GFP^{AD} cells, but did not observe any Env expression (Reviewer Figure 1). Importantly, we do not state that HERV Env expression is required for Tau aggregation or dissemination. Instead, our data suggest that HERV Env

expression can enhance intercellular dissemination of already existing protein aggregates, including Tau.

Reviewer Figure 1. Western blot analysis of HERV Env expression in HEK Tau-GFP^{AD} cells. As controls, HEK cells were transfected with constructs coding for HERV-K, -W, -R and -H Env. Env proteins were detected using monoclonal antibodies raised against respective HERV Envs.

Being aware of the current lack or shortage of suitable in vivo HERV models and given that particularly the pathogenic HERV entities are human-specific, I would nevertheless prefer if corresponding experiments could be performed using as close as possible parameters relevant for the human diseases. This encompasses human primary or iPSC derived neuronal cells (as donors as well as recipients) and not cell lines. In this context I wonder whether only a neuron-to-neuron transfer is relevant and whether glial cells as donors or recipients are also conceivable.

Response: See response above. We agree with the reviewer that it is highly desirable to use primary neurons or glial cells as donors and as recipients for our quantitative cell-based assays. Unfortunately, due to the lengthy procedures required for the generation of donor populations that homogeneously express aggregation-prone reporters such as Tau or NM, produce protein aggregates and express ERV gene products, combined with the delicate nature of primary cells that limits transfection/ transduction, cultivation time, medium exchange and replating, these experiments can only be successful (if at all) if one expects drastic effects of Env expression. Example: Ectopic Tau-GFP expression in 70 % of cells, aggregate induction by transfecting Tau fibrils in 20 % of cells, ectopic Env expression in 70 % of cells = 10 % of this cell population are actual donors if they survive these manipulations. With only 10 % actual donors we unfortunately reach experimental limits. Of note, our induction efficiencies with AD brain homogenate in HEK Tau-GFP cells are usually less than 5 %.

We have, however, now included human primary astrocytes as recipients, clearly demonstrating that expression of Env from two HERV clades in donors strongly increases aggregate induction in those recipients (**Fig. 7i-k**).

Moreover, HERV-K variant HML-2 and HERV-W variant pV14 revealed to be of relevance for a number of human pathologies. So, the MSR-pV14 (GenBank, AF331500) encoding the corresponding HERV-W variant should be considered and tested. Syncytins are rather inappropriate as these are domesticated proteins with physiological functions and to my knowledge no CNS expression was shown so far. Please also consider that Charvet et al.,

2021 showed that the pathogenic form of HERV-W present in multiple sclerosis (AF331500) leads to different oligomerization compared to physiological Syncytin-1.

Response: Thank you for pointing out this interesting HERV-W variant. The protein has recently been published to be secreted as a hexameric protein and thus likely has different interactors and activities as membrane-bound HERV-W (ERVW1) Env Syncytin-1.

*Expression of Env MSRV-pV14 has been linked specifically to MS¹⁶, which is not associated with prion-like spreading of amyloid and thus not focus of our study. The HERV-W Env Syncytin-1 (ERVW1) tested by us is a cell-membrane anchored protein with known receptor and fusogenic activity, characteristics required for intercellular membrane contacts and transmission of viral (and proteopathic) cargo. HERV-W variant pV14 lacks fusogenic activity¹⁶. Importantly, ERVW1 Env is expressed in the CNS¹⁷ and has been found upregulated in MS¹⁷. We therefore find it appropriate to test this HERV-W Env variant. As for the HERV-K, we have used the standard HERV-K subgroup HML-2 K113 consensus Env¹⁸. We now also include data on HERV-H and HERV-R Env (**Supplementary Figure 9a-j**).*

Note, HML-2 Env was recently found to bind CD98HC complexed to β 1 integrin (Steiner et al., 2022), pHERV-W (pV14) was shown to bind TLR4 (Madeira et al., 2016). This must also be considered in the study of appropriate recipient cells.

*Response: As outlined above, we feel that the appropriate HERV-W Env to test is Syncytin-1. As we focus on the role of viral Env in entry receptor binding/membrane fusion, relevant receptors are ASCT-1/2^{19,20}. The interaction of HERV-W variant pV14 with TLR4 induces proinflammatory stimulation of immune cells²¹, which is not scope of this study. To confirm involvement of ACT-1/2 in intercellular aggregate dissemination, we have now included an additional control experiment in which we block Syncytin-1 receptor association by expression of a competing feline ERV Env (RD-114) using the same entry receptor in recipient cells²² (**Supplementary Figure 8c, d**). Competitive receptor blockage is a common method to identify viral entry receptors²³. RD-114 expression in recipients abrogated the Syncytin-1 mediated aggregate induction increase, arguing for an involvement of this viral entry receptor in intercellular protein aggregate spreading. We now also show that a truncated Syncytin-1 with increased cell entry and fusogenic activity increases aggregate induction in recipients (**Supplementary Figure 8e-g**).*

*Unfortunately, the entry receptor for HERV-K HML-2 Env has so far not been identified, but entry of HERV-K Env pseudotyped lentivirus demonstrates its presence on HEK cells²⁴. CD98HC complexed to β 1 integrin has been shown to bind soluble HERV-K Env and induce neurotoxicity, but its role as an entry receptor for this virus has not been established²⁵. In our hands, overexpression of CD98HC in recipients does not increase aggregate induction (and even decreased induction) when donors express HERV-K Env, thus arguing against its role as entry receptor (**Reviewer Figure 2**). We have now added our additional finding that a C-terminally truncated HERV-K Env that facilitates efficient and specific entry into susceptible host cells²⁴ further increases intercellular protein aggregates spreading, supporting our finding that Env/entry receptor interactions are enhancing protein aggregate dissemination (**Supplementary Figure 8h-j**).*

Reviewer Figure 2. CD98 overexpression in recipients does not increase intercellular Tau aggregate spreading when donors express HERV-K Env. **a.** CD98 has been identified as a HERV-K Env interaction partner, but it is unclear if it serves as an HERV-K entry receptor. To test this, donor HEK Tau-GFP^{AD} cells expressing HERV-K Env were cocultured with recipient HEK Tau-FR^{sol} cells expressing or not myc epitope-tagged CD98. Please note that for controls, both donor and recipients were also transfected with empty plasmid. Mock- and HERV-K Env- or CD98-myc transfected cells were used in combination cocultures. **b.** Western blot detection of CD98 expression in recipient cells using anti-myc antibodies. **c.** Quantitative analysis of Tau-FR aggregate induction in recipients following coculture. Statistical analysis was performed with one-way ANOVA with Tukey's multiple comparisons test. Expression of CD98 in recipient cells does not increase aggregate induction when donors express HERV-K Env. Do.: Donor; Re.: Recipient.

There are many more mouse endogenous retroviral elements known such as ETnI, ETnII- α , ETnII- β , ETnII- γ , MusD and IAP (Cipriani et al., 2018), often not listed in databases. Could the authors comment on this fact? Were you able to detect these retroviral elements in your proteome analysis? Were they regulated?

Response: The reviewer raises an interesting question. We have now also performed qPCR on cDNA generated from N2a NM-HA^{agg} cells at early and late passage for ETnI, II, MusD and IAP ORFp. We have not found differences in expression levels of these retroelements between early and late cell passages (Supplementary Figure 3e). We also could not detect them in our proteomes. This is in line with previous studies, demonstrating that retroelements underlie different repression. For example, in mice, disruption of B cell function induces MLV expression but leaves IAP, LINE, SINE unaffected²⁶. Likewise, chemicals have divergent effects on the activation of different retroelements in mice²⁶.

HERVs have multiple times been shown to induce inflammation and immunological responses. Have the authors analysed any inflammatory or immunological markers in this context? Even though it is not the leading hypothesis of protein aggregation in AD and PD, there is certain literature (e.g. Sanchez de Groot and Torrent Burgas, 2015) hypothesizing that inflammation might lead to miss-folding of proteins and therefore increasing protein aggregation. Can inflammation as cause for the protein aggregation be excluded?

Response: We agree with the reviewer that inflammation/ inflammatory cytokines could affect protein aggregation. Importantly, our study does not assess spontaneous protein misfolding but instead focuses on seed-induced protein aggregation between cells. To test if cytokines might play a role here, we have now performed several additional experiments. First, we have exposed recipients to the ultracentrifuged supernatant of late passage N2a NM-HA^{agg} donors. This procedure should pellet EVs and potentially remaining NM aggregates, but should leave the cytokines in solution. This supernatant does not result in aggregate induction in recipients (Supplementary Figure 2a). Second, we tested early and late passage N2a NM-HA^{agg} donors for transcripts of inflammatory cytokines, but did not identify increases in expression levels upon prolonged culture (Supplementary Figure 2b). Third, we

tested if expression of Env of the different tested HERV clades induces a cytokine response in HEK Tau-GFP^{AD} cells, again not showing increased expression (**Supplementary Figure 9k**). Lastly, HERV Env expression also did not induce spontaneous NM-GFP or Tau-FR aggregation (**Supplementary Figures 8a, b; 9i, j**). Together with our extensive evidence that receptor- ligand interactions and fusogenic Env activity are required for aggregate induction, this argues that cytokines in our system do not play an essential role in intercellular aggregate induction.

Regarding your statistics, none of your datasets were tested for normal distribution. However, all of your statistical test assume that your datasets are normally distributed. Although mostly showing low numbers of replicates, the Shapiro-Wilk test should be able to test your datasets for normality. If you have performed a test for normal distribution, it should be described in your material and method section, if not please revise your manuscript accordingly. Furthermore, I do not really understand why you are switching from one-way ANOVA with Bonferroni post-test to Students t-test back and forth. The manuscript would benefit from a more clear-cut way of performing statistics (and actually multiple students t-testing is not the best way). GraphPad 6 might not be able to do that, but Bonferroni is also a rather weak statistical post hoc test. Sidak or Tukey would be more powerful.

Response: We apologize for the inaccurate use of statistical tests in several panels. We have revised our statistics accordingly. We have now tested for normality and chosen parametric tests where appropriate. For multiple pairwise comparison of more than 2 groups, we have now performed one-way ANOVA with Tukey's multiple comparisons tests. When comparing selected pairs of means, with the selection based on experimental design, we have used Bonferroni's post hoc test. Further, for pairwise comparisons of multiple treatment groups with a single control group, we have used one-way ANOVA with Dunnett's post hoc test. In several instances, our group sizes are very small (n=3). The use of parametric versus non-parametric tests for very small samples sizes is controversially discussed. Nonparametric tests can have limited to no power for small samples sizes, in which cases the use of parametric test has been suggested²⁷⁻³⁰.

In several instances, our group sizes were too small to confirm normal distribution and the statistical power of non-parametric tests was sometimes too low to achieve significance. This is the case for data shown in Figs. 1c, 2a, 4i, 5j, 6h, 6n, 6o, 6p and 7k and Suppl. Figs. 1c, e, 4c, 4e, 4f and 7c. Of note, here individual groups are never overlapping. Importantly, when using identical experimental settings (such as cocultures) with comparable to bigger sample sizes, the Shapiro-Wilk tests confirms normal distribution of data (e.g. Fig. 1e, 3d, 3k, 5i, 6c, 6d, 6i, 6l, 7d and Suppl. Figs. 4b, 4h, 7d, 8d, 8g, 8j, 9d and 9h). We thus assume normal distribution and used ANOVA with appropriate multiple comparison tests. We have revised the statistics part in the Materials and Methods section.

Could the authors comment on the fact that they often switch between normalized and non-normalized data regarding the “% of recipients”. I think it is rather confusing the reader.

*Response: The reviewer is referring to the fact that we sometimes normalize aggregate induction rates to rates observed in controls, rather than showing non-normalized data. We would like to keep this way of normalization whenever we are comparing different treatments against controls. This is especially true for dose response curves (**Fig. 4**) for which such normalization is standard. In most cases we show non-normalized data for readers to appreciate actual induction rates.*

Minor comments:

Legend Fig. 1c is inconsistent with graph labelling: “c. Percentage of recipient cells with induced NM-GFP aggregates upon coculture with donor N2a NM-HAagg cells P6 and P17.”
Response: We apologize for this confusion and have now correctly labelled the passage numbers.

Fig. 3b: awkward presentation of lowered levels upon siRNA (“fold change over control”) use instead bars also for control levels so that effects can intuitively been recognized. This is also true for most of the data shown in Fig. 3.

*Response: We have now changed the presentation to also include the controls in **Fig. 3b, f and i**. We changed the labelling of the y axis to “Relative mRNA levels”.*

Fig. 3d: Why are there two graphs labeled with cocultures shown? Would the right one not be EVs?

Response: We apologize for missing labelling. Graphs show results of cocultures following Env or Gag silencing. We have now adequately labelled the two graphs.

Legend Fig. 3: “were transfected”

Response: This was corrected.

Fig. 4: The figure legend of Fig. 4 might still contain some writing artefacts “(f????)”?

Response: This was corrected.

Fig. 5: Awkward demonstration of receptor downregulation in Fig. 5b., which is also true for 5c and 5d; avoid “normalized to control”.

*Response: We now show relative changes of the respective receptor mRNAs following siRNA-mediated knock-down compared to controls in **Fig. 5b**. For **Fig. 5c and d**, we prefer to show normalized data to follow our standard procedure when comparing different treatment of the same cells (please compare to **Figure 4 a-f**).*

Fig. 6: Figures 6c-e should best be combined in one graph, Figures 6i and m and n should also be combined.

*Response: The reviewer suggests to combine graphs displayed in **Fig 6c-e** or **i-n** for simplicity. Unfortunately, this cannot be done, as for each panel, a different amount of total plasmid DNA was transfected. To account for potential effects of DNA transfection and because experiments were independent, we had to keep the graphs separately.*

Reviewer #2 (Remarks to the Author):

Studies like Liu et al. deserve full attention and support. Why? Over recent decades, there has been great progress in understanding the molecular details of familial forms of neurodegeneration and dementia. Yet, spontaneous, not inherited, cases represent the epidemiological bulk. Their mechanisms are only rudimentary known when compared to our understanding of the familial neurodegeneration. The burgeoning field of neuroinflammation has recently begun to correct this imbalance. Previous work of Vorberg’s group indicated that viruses, actually infectious agents, can support the spread of aggregates even without induction of inflammation. In the current study, they continue the line by discovering that and explaining how the reactivated endogenous retroviruses can do this as well. To facilitate the proper perception of the importance of these findings by general readership I would recommend to simplify the title a little bit, for example “Reactivated endogenous retroviruses promote aggregate spreading” or similar.

Response: We appreciate the reviewer's strong support of our findings. We thank the reviewer for the simplified title, which we have now adapted.

While reading the manuscript, I was looking for experimental evidence in support of some kind of signaling-related mechanism(s) and could not find any. On the contrary, I became convinced of the structural role of reactivated retroviruses as drivers and vehicles that mediate the spread of seeds of aggregation in the cell culture models used by the authors. The clarification of the data presented Fig.2h might be helpful in consolidating this impression. Specifically, how does the cumulative aggregate-spreading “potency” of the fractions 2 to 6 compare to the original potency of the conditioned medium used for pelleting and fractionation? If fractions can do the job only weakly, is there a possibility of additional (soluble) factors that are lost during pelleting and/or fractionation? Regardless of these optional considerations, I found the fractionation experiments very important in view of future work. It is quite obvious that structural biologists will be keen to analyze the structural characteristics of proteopathic seeds transported in the donor cell-generated extracellular vesicles in this experimental model.

*Response: We thank the reviewer for his/her enthusiasm for our findings. The reviewer wonders how the seeding activity of the density gradient fractions compares to the activity of the conditioned medium. Please note that the design for the different experiments unfortunately precludes direct comparison of seeding activity. In the case of conditioned medium, no concentration of EVs by ultracentrifugation has been performed. In the case of the density gradient, EVs and virus derived from a very large volume of conditioned medium were first concentrated by ultracentrifugation and subsequently fractionated on a gradient. The experiment in **Fig. 2h** only intends to demonstrate that the highest seeding activity is present in fractions 2-6, while virions can be separated in fraction 10. To demonstrate that no seeding activity remains in the soluble fraction after ultracentrifugation, we have now also tested this supernatant for seeding efficiency, demonstrating that seeding activity can be fully pelleted (**Supplementary Figure 2a**).*

What I could not understand is the fact that only a certain fraction, typically 30-40 %, of recipient cells converts their soluble aggregation-prone proteins to real aggregates. Does the vulnerability depend on the cell cycle status of the exposed recipient cells? What are other possible explanations? Could the authors discuss these possibilities in the manuscript?

*Response: The reviewer raises an interesting question why only a certain number of recipient cells converts the soluble protein into aggregates. This somehow resembles infection of cells with prions, proteinaceous infectious particles composed of misfolded prion protein³¹. The reviewer raises the question if induction rates could be dependent on cell cycle phases. We have now tested this by determining the number of cells expressing Cyclin B1, (a cell cycle marker highly expressed during transition from G2 to mitosis) with and without induced aggregates. Interestingly, only a weak effect of cell cycle was observed. However, we cannot exactly define in which cell cycle phase induction is most efficient, as first aggregate induction can be observed 6 hours post exposure to donors or donor EVs. Please note, however, that cell cycle progression is not necessary for NM aggregate induction, as such aggregates can be induced in postmitotic primary neurons (**Fig. 1g**) and neurons in cerebellar brain slices³². As mitotic activity is no requirement for NM aggregate induction and this is not scope of our study, we only include the data for the reviewer (**Reviewer Figure 3**).*

Reviewer Figure 3. Low effect of cell cycle on intercellular protein aggregate induction. *a.* To test the effect of cell cycle on intercellular protein aggregate induction, donor N2a NM-HA^{agg} cells (LP) and recipient N2a NM-GFP^{sol} cells were cocultured for 16 h and subsequently stained for Cyclin B1 (red), a key regulator of mitotic entry. The expression of Cyclin B1 is cell cycle dependent and is highest at the G₂/mitosis transition. *b.* Close-ups of cells with and without increased Cyclin B1. Recipient cells positive (*, +) and negative (arrowhead, °) for Cyclin B1 are shown. *c.* The percentage of recipient cells with induced NM-GFP aggregates expressing Cyclin B1 was slightly increased compared to the percentage of recipients not expressing Cyclin B1. Statistical analysis was performed using t-test (n=3 independent cocultures).

Why the XPR1-transfected HEK293T cells reach the similar efficiency as N2a cells (Fig. 5h)? Does it mean that the transfection efficiency reaches 100%? How efficient would be the conversion of recipient cells with a lower transfection rate?

Response: We apologize for not explaining in more detail how the XPR1-expressing HEK cells were generated. For this, recipients were transfected with XPR1-HA Piggyback and transposase vectors. XPR1-HA positive cells were subsequently enriched by puromycin selection and single cell cloning. We have now analyzed our XPR1-HA expressing recipients for transgene expression and demonstrate that more than 80 % of cells expresses the tagged receptor (**Supplementary Figure 7c**). We have included information on the Piggyback-generated cell line in the Materials and Methods section.

Although the conversion does not happen in all cells, I do strongly support the simplified rendering of the experimental setups as provided by the authors in almost all figures. These schemes make following the text and data easy, despite the complexity of the performed experiments. Also more generally, the flow of the text and the argumentation are well structured and clear and a pleasure to read (except of the last paragraph of the Discussion).

Response: Thank you for your compliments on our text and experimental schemes.

Several minor issues:

Fig. 1h vs. 1j: mRNAs of gag and, especially, env in cells are upregulated much more than the respective proteins. Is there a problem with translation of the viral mRNAs in the system?

Response: The reviewer refers to differences in mRNA and respective protein levels. Multiple reasons can account for the observed differences in RNA and protein levels. Generally, RNA levels do not necessarily reflect protein levels³³. Furthermore, MLV retrovirus produce different types of RNA. The full-length RNA serves as genome for virions as well as a transcript for translation of Gag and Pol. It is further spliced into subgenomic RNA that is translated into Env^{34, 35}. The ratios of genomic and subgenomic viral RNAs present in our cells are unknown. Interestingly, for virus production, all ORFs are initially transcribed as one RNA, still different levels of viral proteins are produced depending on their requirements for virion assembly³⁶.

To clarify proviral structure and transcripts, we have now included **Supplementary Figure 3**.

Fig. 1g: please indicate the gag processing products MA and NC as they are mentioned in the text (lane 160).

*Response: The anti-Gag antibody from Abcam is derived against the capsid part of the Gag polyprotein and does not detect matrix (MA) and nucleocapsid (NC). We have now deleted MA and NC from the blot. For better illustration, we have now included **Supplementary Figure 3**, which shows cleavage products of Env and Gag.*

Fig. 4c: swap the upper and lower schemes to make the alignment with the respective curves in Fig. 4d more natural.

Response: This has been done.

Fig. 4 legend, lane 1254: (f????) needs to be corrected.

Response: This has been corrected.

Fig. 5b: delete XPR1/mCat-1 from the Y-axis label, otherwise confusing.

Response: This has been revised.

Fig. 5e: provide continuous sequences of the loops.

*Response: We have now included the continuous sequences of the loops in **Supplementary Figure 7b**.*

Fig. 6c-e: set the same Y-axis scale for the ease of comparison.

Response: Y-axes have been adjusted.

Fig. 6l-m: as above.

Response: Y-axes have been adjusted.

I strongly support publication of this study.

Response: We thank the reviewer for his/her strong support.

Reviewer #3 (Remarks to the Author):

In this manuscript, the authors have explored that up-regulation of the murine endogenous retroviral (MLV) proteins env and gag results in intracellular aggregate induction as well as aggregate spreading. Targeting MLV maturation using protease inhibitors impaired intracellular protein aggregation. Moreover, overexpression of specific HERV glycoproteins increased intercellular protein aggregation.

However, the study unfortunately lacks both the mechanism of how MLV gag/pol and env get increased

*Response: The reviewer points out that we have not clarified which events lead to the spontaneous formation of active MLV in our N2a cells. Spontaneous derepression of ERVs and production of active viral particles in cell culture has been observed many times over the last decades¹⁻⁶. While extremely interesting, identifying the so-far unknown mechanisms that lead to this well-accepted phenomenon is beyond the scope of this study. However, it has been reported that endogenous MLV can become upregulated upon hypomethylation^{37, 38}. Indeed, we demonstrate that methylating or demethylating epigenetic drugs affect the expression of MLV and intercellular protein aggregate spreading (**Fig. 3f-k**).*

To discuss in more detail the events that likely lead to virus production, we have now included a paragraph in the discussion section, clarifying that formation of active polytrophic MLV is likely the result of reactivation of epigenetically silenced MLV followed by recombination events, as has been observed in mouse models (page 18).

as well as the mechanism of how MLV gag/pol and env lead to the induction of intracellular aggregates.

*Response: The reviewer suggests that expression of MLV gag/pol leads to the induction of intracellular protein aggregates. We believe this is a misunderstanding. ERV gene products do not result in spontaneous protein aggregate formation, which we now show in **Supplementary Figures 8a,b and 9 i, j**. NM protein aggregates in N2a cells had been induced by recombinant NM fibrils and donor cells persistently replicate induced aggregates since then³⁹. NM aggregates also propagate in MLV-negative HEK cells. Reconstitution of MLV expression in another cell system does not induce NM or Tau aggregation (**Fig. 6**). Instead, MLV gene products increase the cell-to-cell spreading of protein aggregates by facilitating cell-to-cell or EV-to-cell contacts through viral receptor-ligand interactions. We have provided conclusive evidence that this is the case (reconstitution experiments of MLV and cognate receptor in HEK cells, impairment of aggregate induction by anti-Env antibodies, anti-MLV siRNA, promoter methylation, viral protease inhibition). Our comprehensive analyses convincingly clarify how MLV gene products affect protein aggregate spreading.*

Thus, the data and experiments presented in this study require substantial additional changes before publication.

Major Comments:

1) Since only gag and env of MLV were found upregulated in the proteomic analyses and these were then also examined throughout the manuscript, the current manuscript title is misleading as it implicates that all ERVs affect intracellular aggregates. This should be made more specific.

*Response: In the revised manuscript we have now shown that gene products of murine MLV and envelope proteins from four different classes of human ERVs (HERV-W, -K, -R and -H) increase intercellular protein aggregate spreading, using both Tau and NM cell-based assays (**Fig. 7; Supplementary Figure 8, 9a-h**). We therefore believe it is valid to argue that endogenous retroviruses are capable of promoting protein aggregate spreading.*

2) To show that only MLV gag/pol and env are affecting intracellular NM aggregation and aggregate spreading it would be necessary to also investigate the cells on transcript levels (RNAseq) beside the proteomics approach that was conducted. This would show if other transcripts not coding for env or gag would also play roles in the observed findings. This is not unlikely, since the vast majority of HERV elements in our genome consist of solitary LTRs and mutant HERV proviruses. In addition, it could also help to provide a better insight into the mechanisms that affect intracellular NM aggregation and aggregate spreading.

Response: The reviewer suggests to use RNAseq to identify additional factors that might contribute to the intercellular spreading of protein aggregates. We apologize we accidentally did not include the table of our proteomic analysis. Data have now been included (Supplementary Table 1). Proteomic analyses of cell lysate and EVs of early and late passage donors demonstrate that host proteins other than MLV proteins are differentially expressed. While it will be interesting to study if any of these proteins have any effect on the spreading, this is unfortunately beyond the scope of this study.

We have provided conclusive evidence that MLV gene products are fully sufficient to increase intercellular spreading of protein aggregation. First of all, we have demonstrated that increased spreading is crucially dependent on MLV Env-receptor interactions and can be strongly reduced by receptor polymorphisms, anti-Env antibodies and antivirals that impair Env and Gag maturation. We have further demonstrated that reconstitution of HEK donors with combinations of MLV gene products can increase aggregate induction in recipients. Finally, we demonstrated that simple overexpression of (now) four different HERV Env is sufficient to increase intercellular aggregate spreading. Combined, our data reveal that the tested viral gene products are sufficient in our cell models to drive intercellular aggregate spreading. If (and which) other cellular processes might be triggered by viral gene products and might contribute to the spreading is beyond the scope of this study.

*We agree with the reviewer that it is interesting to know if other retroelements become upregulated. We have therefore performed additional qPCR analysis on donor cells of early and late passage, revealing no increase in transcript levels of representative retroelements (**Supplementary Figure 3e**). Please note that variable methylation of retroelements (and thus activation) is well documented ⁴⁰.*

*We have now also tested for upregulation of a panel of inflammatory cytokines, showing no cytokine upregulation upon prolonged N2a NM-HA^{qsg} cell culture (**Supplementary Figure 2b**).*

3) The authors show that there is an increase in MLV gag and env proteins in specific neuroblastoma Sup35 NM mouse model. However, several things are unclear concerning the experiments in which MLV gag or env are downregulated by siRNAs or reactivated by epigenetic drugs to show that they are related to intracellular NM aggregation:

a) After knockdown with the specific MLV siRNAs, RNA (Fig 3B) and protein (Fig3 C) levels behave very differently. While on the RNA level only a 50 % knockdown with an MLV env siRNA can be detected, on the protein level a much more efficient knockdown can be detected. For MLV gag it behaves the opposite. Here a more efficient knockdown is detectable on RNA level whereas on Protein level nearly no knockdown is visible (considering that also GAPDH is affected here).

How can this be explained? In addition, the WB's (Fig 3C, G, J) must be quantified so that a comparison is easy and changes in GAPDH levels are considered.

*Response: We have now quantified normalized protein levels of WB shown in **Fig. 3c, g, j** to more easily compare them with the respective mRNA knock-downs.*

We would like to stress that multiple reasons can account for the observed differences in RNA and protein levels.

Generally, RNA levels do not necessarily reflect protein levels ³³. As half-lives of proteins can differ, siRNA mediated knock-down can affect already available protein pools differently ⁴¹.

Further, MLV retrovirus produce different types of RNA. The full-length RNA serves as genome for virions as well as a transcript for translation of Gag and Pol. It is further spliced into a subgenomic RNA that is translated into Env ^{34, 35}. The ratio of genomic versus subgenomic viral RNAs present in our cells is unknown. Interestingly, for virus production, all ORFs are initially transcribed as one RNA, still different levels of viral proteins can be produced depending on their requirements for virion assembly ³⁶.

*To clarify proviral structure and transcripts, we have now included **Supplementary Figure 3a-d**.*

b) The authors justify the weak knockdown via existing multicopy ERVs. Which MLVs copies are exactly targeted by the siRNAs? The qPCR Primers are targeting which copies?

Response: We apologize for not explaining primers and probes in more detail.

MLV exist as multicopy endogenous proviruses in the murine genome. Common inbred mouse lines carry one to few ecotropic⁴² and more than 110 non-ecotropic MLVs, including approx. 47 polytropic proviruses⁴³. As MLV proviruses are polymorphic between mouse lines and new recombinants arise due to intersubgroup recombinations⁴⁴, the exact loci involved are usually unknown, also here.

Our mass spec analyses revealed that MLV Env with a polytropic host range (P10404) was highest in late passage N2a cells and EVs.

*As for siRNA used in **Fig. 3c**, three custom-designed siRNAs targeting AY219544.2 (P10404: 100 % similarity to AAO37244.2 Envelope protein encoded by AY219544.2 nucleotide sequence) were from Thermofisher. Due to the short sequence, we assessed sequence similarity to a library of identified MLV proviruses⁴⁵. siRNAs bound with maximum of 1 mismatch to non-ecotropic X/P-MLV sequences. For ecotropic sequences, 3-10 mismatches existed. We conclude the siRNAs likely silence X/P-MLV env. To identify env transcripts, Taqman assays were designed by the company based on the provided genomic sequence AY219544.2 (nucleotides 2786-3298). The exact sequences of the primers/probes are company IP and have not been shared with us. An NCBI nucleotide blast search using the respective env nucleotide region (amplicon) reveals 100 hits with sequence similarity of at least 99,6 %. Where reported, those code for polytropic Env. We conclude that this Taqman assay is capable of detecting polymorphic polytropic env transcripts. We have now included a section on the Taqman assay in the methods section. We also included a sentence in the figure legend explaining that the Taqman env assay is designed to detect polytropic env.*

*Infectious P-MLV viruses carry ecotropic backbones with P-MLV related changes in the 3' pol and 5' env region⁴⁶. Three designed siRNAs targeting gag were aligned to a library of known MLV proviruses⁴⁵. siRNA 1 bound without mismatch to all ecotropic and non-ecotropic X/P-MLV sequences. siRNA 2 bound without mismatches to ecotropic gag, up to 4 mismatches to polytropic. Due to many mismatches, binding to X-MLV is unlikely. siRNA 3 bound to ecotropic and non-ecotropic X/P-MLV gag with a maximum of 1 mismatch. To identify gag transcripts, genomic sequence J01998.1 coding for ecotropic MLV, nucleotides 756-1219, was provided for Taqman assay design. Due to extensive gag sequence overlap, this region shares at least 93 % similarity with approx. 90 ecotropic, xenotropic and polytropic MLV gag sequences according to NCBI nucleotide blast. We conclude that this Taqman assay is likely capable of detecting eco- and non-ecotropic MLV gag transcripts. We conclude that the combination of siRNAs used was designed to silence both eco- and non-ecotropic MLV gag transcripts. We have now included this information in the materials and methods section (**pages 35-36**).*

c) How do the authors exclude off target effects by using siRNAs against multiple copies of MLV gag or env genes? It would be necessary to at least show that MLV gag protein levels are unaffected upon siRNA treatment against MLV env and vice versa for the siRNAs against MLV gag.

*Response: The reviewer rightfully points out that multiple copies of MLV provirus exist in the murine genome and asks how we can demonstrate that siRNA directed against env does not affect gag/pol expression and vice versa. In fact, we cannot. This is due to the genomic structure of the provirus and its genomic and subgenomic transcripts. A genomic mRNA is transcribed from the provirus, coding for gag/pol and env. Splicing of this RNA produces a subgenomic mRNA only coding for env. Consequently, any siRNA targeting gag/pol or env will target the genomic mRNA and ultimately expression of gag/pol and env. To explain in more detail the genomic structure of the provirus and its transcripts, we have now included **Supplementary Figure 3a-d**.*

d) The WB for gag in Fig 3C, G and J is cut. Therefore, it is not possible to see the different gag bands. How do they behave? Are they also regulated by the siRNAs and epigenetic drugs? This would be interesting since they are strongly induced in Fig 1K.

*Response: We have now included the uncut Western blots for **Figure 3c and j**, demonstrating that siRNA knock-down and epigenetic drugs similarly affect the levels of full-length and processed Gag. Unfortunately, we cannot provide the Western blot analysis for processed Gag in **Fig. 3g**, as we had cut this blot for simultaneous GAPDH detection. However, based on the fact that we observe a correlation between full-length and processed Gag levels in **c and j**, we assume a similar trend for **Fig. 3g**.*

4) The authors can show that that upregulated MLVs affecting NM aggregate induction belong to the X/P-MLVs. These data suggest that these MLVs are the ones driving the observed effects. Therefore it would be interesting to target only X/P-MLVs specifically using siRNAs or other approaches (e.g. CRISPRi).

*Response: The reviewer suggests to use CRISPR to abolish expression of X/P-MLVs in our N2a cell model. Unfortunately, this would be technically extremely challenging, as the murine genome consists of dozens of polymorphic X- and P-MLV proviruses⁴⁵. Replication-competent P-MLV arise from recombination events between different MLV subgroups, complicating matters⁴⁷. Finally, our cells also produce infectious particles that likely result in additional genomic integrations. Most importantly, we have experimentally proven that the receptor for our virus is XPR1. **This defines the MLV as subgroup X/P**⁴⁸. Expression of this receptor turns resistant HEK cells into recipients (**Fig. 5h**). As xenotropic viruses are unable to infect murine cells⁴⁹, this identifies our MLV virus as polytropic.*

*To clarify, we have now pointed out the high number of MLV proviruses in the murine genome and that recombination events between different MLV subgroups result in the formation of active virus with polytropic host range (**page 5**). We have now also discussed the events that likely result in virion production in more detail in the discussion section (**page 18**).*

5) The authors show that expression of HERV-W env Syn-1 as well as HERV-K env K113 can increase the intracellular protein aggregation and spreading. To show that these effects are rally HERV-W and HERV-K env specific the authors should also include an HERV derived envelope from a HERV group, which has not been associated with neuropathy.

*Response: The reviewer raises the interesting point that expression of some HERV envelope proteins might not be neuropathic. We believe that at the current state of research, one cannot define certain HERVs as non-pathogenic when aberrantly expressed. Excitingly, recent advances in retroelement detection in RNAseq data has now demonstrated an upregulation of different classes of HERVs also in neurodegenerative diseases⁸⁻¹⁵. A cause-correlation of retroelement expression for disease initiation/progression has yet to be established. We hypothesized that HERV Env proteins that are still able to mediate contact with their cognate receptors and/or have fusogenic potential can contribute to intercellular aggregate spreading. We have shown this for HERV-K and -W Env and now also included HERV-R and -H Env (**Supplementary Figure 9a-h**). All HERV Envelope proteins increased protein aggregate spreading. Further, we now also included human primary astrocytes as recipients, demonstrating that tested HERV-W and HERV-K Env can also mediate spreading of Tau aggregates to this cell type (**Figure 7i-k**).*

Minor Comments:

1) Supplementary Table 1 was unfortunately not included or accessible.

*Response: We apologize for not providing the **Supplementary Table 1**, which has now been included.*

2) Why does the env expression behave so different in the control cells of Figure 3G compared to the control cells of Figure 3C and J? The lower band here is much weaker, which then naturally entails a rise to Aza and Dec treatment. When comparing the increase to the other control cells, the increase in MLV Gag and Env would be more or less the same after Aza and Dec treatment. Again, quantification of the WBs is necessary for comparison. Besides, the increase in aggregates Fig 3H is significantly higher after Aza treatment than after Dec treatment. How can this be explained? Could this be due to other, non MLV Env/Gag effects?

*Response: The reviewer comments on the different maturation states of the Env protein in experiments **Fig. 3c and j** versus **Fig. 3g**. Shown are the MLV Envelope protein precursor which migrates ~85 kD, whereas after SU/TM cleavage, the SU migrates ~70 kD⁵⁰. Please note that the cells in panels **c, j** are late passage cells (depicted above the Western blot), while cells in **g** are early passage.*

One difference in these cell populations is that expression of MLVs is very low at early passage and strongly increased in high passage cells. Differences are likely related to changes in maturation/degradation states of newly translated versus existing Env pools. Please note that pretreatment times differ markedly, making direct comparisons of Env protein maturation difficult.

*In early passage cells (**Fig. 3g**, no treatment) we observe a more prominent Env precursor at ~85 kD, potentially because at low expression it has not been properly processed. When we increase Env translation pharmacologically, we detect increased newly translated, so far non-processed Env precursor.*

*In late passage cells (**Fig 3c, j**) the processed SU domain at 70 kD band is more prominent without cell treatment. Genetical or pharmacological silencing of Env expression is expected to change the rate of newly translated precursor (background expression) to processed Env (degradation). We have now clarified this in the figure legend. We have also quantified levels of Env and Gag, as requested.*

3) In Fig 4B it would be easier to follow the different protease inhibitors if a color code would be used.

*Response: Thank you for this suggestion. **Fig. 4b** has now been color-coded.*

4) How do gag and env levels behave in Fig 5d and i?

*Response: We are afraid we do not understand this question. For these experiments in **Fig. 5d**, only recipient cells were transfected, so donors were not treated. As we have not manipulated donors, we do not anticipate changes.*

5) Line 289-290: "Highest induction rates in cocultures were observed when donor cells were also transfected with retroviral transfer vector (TV) (Fig 6e)." The induction in Fig 6e compared to Fig 6d is not increased if one compares +gag/pol, +env (Fig6D) and +TV, +gag/pol, +env (Fig6E). In both samples values are approx. 22-24.

*Response: The reviewer compares results displayed in **Fig. 6d and e**. Unfortunately, a direct comparison here is not possible, as these are independent experiments in which we used different combinations of plasmids and different total amounts of plasmids (due to transfection of 2 or 3 plasmids).*

References

1. Alais, S. *et al.* Mouse neuroblastoma cells release prion infectivity associated with exosomal vesicles. *Biology of the cell / under the auspices of the European Cell Biology Organization* **100**, 603-615 (2008).
2. Aaronson, S.A., Hartley, J.W. & Todaro, G.J. Mouse leukemia virus: "spontaneous" release by mouse embryo cells after long-term in vitro cultivation. *Proc Natl Acad Sci U S A* **64**, 87-94 (1969).
3. Pothlichet, J., Heidmann, T. & Mangeney, M. A recombinant endogenous retrovirus amplified in a mouse neuroblastoma is involved in tumor growth in vivo. *International journal of cancer. Journal international du cancer* **119**, 815-822 (2006).
4. Ottina, E. *et al.* Restoration of Endogenous Retrovirus Infectivity Impacts Mouse Cancer Models. *Cancer Immunol Res* **6**, 1292-1300 (2018).
5. Rasheed, S. *et al.* Spontaneous release of endogenous ecotropic type C virus from rat embryo cultures. *J Virol* **18**, 799-803 (1976).
6. Lieber, M.M. & Todaro, G.J. Spontaneous and induced production of endogenous type-C RNA virus from a clonal line of spontaneously transformed BALB-3T3. *International journal of cancer. Journal international du cancer* **11**, 616-627 (1973).
7. Liu, S. *et al.* Highly efficient intercellular spreading of protein misfolding mediated by viral ligand-receptor interactions. *Nat Commun* **12**, 5739 (2021).
8. Douville, R., Liu, J., Rothstein, J. & Nath, A. Identification of active loci of a human endogenous retrovirus in neurons of patients with amyotrophic lateral sclerosis. *Ann Neurol* **69**, 141-151 (2011).
9. Douville, R.N. & Nath, A. Human Endogenous Retrovirus-K and TDP-43 Expression Bridges ALS and HIV Neuropathology. *Front Microbiol* **8**, 1986 (2017).
10. Manghera, M., Ferguson, J. & Douville, R. Endogenous retrovirus-K and nervous system diseases. *Curr Neurol Neurosci Rep* **14**, 488 (2014).
11. Arru, G. *et al.* Humoral immunity response to human endogenous retroviruses K/W differentiates between amyotrophic lateral sclerosis and other neurological diseases. *Eur J Neurol* **25**, 1076-e1084 (2018).
12. Guo, C. *et al.* Tau Activates Transposable Elements in Alzheimer's Disease. *Cell Rep* **23**, 2874-2880 (2018).
13. Frost, B., Hemberg, M., Lewis, J. & Feany, M.B. Tau promotes neurodegeneration through global chromatin relaxation. *Nat Neurosci* **17**, 357-366 (2014).
14. Sun, W., Samimi, H., Gamez, M., Zare, H. & Frost, B. Pathogenic tau-induced piRNA depletion promotes neuronal death through transposable element dysregulation in neurodegenerative tauopathies. *Nat Neurosci* **21**, 1038-1048 (2018).
15. Jeong, B.H., Lee, Y.J., Carp, R.I. & Kim, Y.S. The prevalence of human endogenous retroviruses in cerebrospinal fluids from patients with sporadic Creutzfeldt-Jakob disease. *J Clin Virol* **47**, 136-142 (2010).
16. Charvet, B. *et al.* Human Endogenous Retrovirus Type W Envelope from Multiple Sclerosis Demyelinating Lesions Shows Unique Solubility and Antigenic Characteristics. *Virol Sin* **36**, 1006-1026 (2021).
17. Schmitt, K., Reichrath, J., Roesch, A., Meese, E. & Mayer, J. Transcriptional profiling of human endogenous retrovirus group HERV-K(HML-2) loci in melanoma. *Genome Biol Evol* **5**, 307-328 (2013).
18. Hanke, K. *et al.* Reconstitution of the ancestral glycoprotein of human endogenous retrovirus k and modulation of its functional activity by truncation of the cytoplasmic domain. *J Virol* **83**, 12790-12800 (2009).

19. Marin, M., Lavillette, D., Kelly, S.M. & Kabat, D. N-linked glycosylation and sequence changes in a critical negative control region of the ASCT1 and ASCT2 neutral amino acid transporters determine their retroviral receptor functions. *J Virol* **77**, 2936-2945 (2003).
20. Lavillette, D. *et al.* The envelope glycoprotein of human endogenous retrovirus type W uses a divergent family of amino acid transporters/cell surface receptors. *J Virol* **76**, 6442-6452 (2002).
21. Madeira, A. *et al.* MSR/V envelope protein is a potent, endogenous and pathogenic agonist of human toll-like receptor 4: Relevance of GNBAC1 in multiple sclerosis treatment. *J Neuroimmunol* **291**, 29-38 (2016).
22. Shimode, S., Nakaoka, R., Shogen, H. & Miyazawa, T. Characterization of feline ASCT1 and ASCT2 as RD-114 virus receptor. *J Gen Virol* **94**, 1608-1612 (2013).
23. Blond, J.L. *et al.* An envelope glycoprotein of the human endogenous retrovirus HERV-W is expressed in the human placenta and fuses cells expressing the type D mammalian retrovirus receptor. *J Virol* **74**, 3321-3329 (2000).
24. Kramer, P. *et al.* The human endogenous retrovirus K(HML-2) has a broad envelope-mediated cellular tropism and is prone to inhibition at a post-entry, pre-integration step. *Virology* **487**, 121-128 (2016).
25. Steiner, J.P. *et al.* Human Endogenous Retrovirus K Envelope in Spinal Fluid of Amyotrophic Lateral Sclerosis Is Toxic. *Ann Neurol* **92**, 545-561 (2022).
26. Sankowski, R. *et al.* Endogenous retroviruses are associated with hippocampus-based memory impairment. *Proc Natl Acad Sci U S A* **116**, 25982-25990 (2019).
27. Winter, J.C.F. Using the Student's t-test with extremely small sample sizes. *Practical Assessment, Research & Evaluation* **18** (2013).
28. Janusonis, S. Comparing two small samples with an unstable, treatment-independent baseline. *J Neurosci Methods* **179**, 173-178 (2009).
29. Zimmermann, D.W., Zumbo, D. B. Parametric alternatives to the student t test under violation of normality and homogeneity of variance. *Perceptual and Motor Skills* **74**, 835-844 (1992).
30. Sawilowsky, S.S., Hillman, S. B. Power of the independent samples t test under a prevalent psychometric measure distribution. *Journal of Consulting and Clinical Psychology* **60**, 240-243 (1993).
31. Vorberg, I., Raines, A., Story, B. & Priola, S.A. Susceptibility of common fibroblast cell lines to transmissible spongiform encephalopathy agents. *J Infect Dis* **189**, 431-439 (2004).
32. Hofmann, J.P. *et al.* Cell-to-cell propagation of infectious cytosolic protein aggregates. *Proc Natl Acad Sci U S A* **110**, 5951-5956 (2013).
33. Gygi, S.P., Rochon, Y., Franza, B.R. & Aebersold, R. Correlation between protein and mRNA abundance in yeast. *Mol Cell Biol* **19**, 1720-1730 (1999).
34. Murphy, E.C., Jr., Campos, D., 3rd & Arlinghaus, R.B. Cell-free synthesis of Rauscher murine leukemia virus "gag" and "env" gene products from separate cellular mRNA species. *Virology* **93**, 293-302 (1979).
35. Zaane, D.V., Gielkens, A.L., Hesselink, W.G. & Bloemers, H.P. Identification of Rauscher murine leukemia virus-specific mRNAs for the synthesis of gag- and env-gene products. *Proc Natl Acad Sci U S A* **74**, 1855-1859 (1977).
36. Shehu-Xhilaga, M., Crowe, S.M. & Mak, J. Maintenance of the Gag/Gag-Pol ratio is important for human immunodeficiency virus type 1 RNA dimerization and viral infectivity. *J Virol* **75**, 1834-1841 (2001).
37. Gaudet, F. *et al.* Induction of tumors in mice by genomic hypomethylation. *Science* **300**, 489-492 (2003).

38. Rowe, H.M. & Trono, D. Dynamic control of endogenous retroviruses during development. *Virology* **411**, 273-287 (2011).
39. Krammer, C. *et al.* The yeast Sup35NM domain propagates as a prion in mammalian cells. *Proc Natl Acad Sci U S A* **106**, 462-467 (2009).
40. Elmer, J.L. *et al.* Genomic properties of variably methylated retrotransposons in mouse. *Mob DNA* **12**, 6 (2021).
41. Gavrillov, K. *et al.* Enhancing potency of siRNA targeting fusion genes by optimization outside of target sequence. *Proc Natl Acad Sci U S A* **112**, E6597-6605 (2015).
42. Kozak, C.A. & O'Neill, R.R. Diverse wild mouse origins of xenotropic, mink cell focus-forming, and two types of ecotropic proviral genes. *J Virol* **61**, 3082-3088 (1987).
43. Frankel, W.N., Stoye, J.P., Taylor, B.A. & Coffin, J.M. A linkage map of endogenous murine leukemia proviruses. *Genetics* **124**, 221-236 (1990).
44. Evans, L.H. & Cloyd, M.W. Friend and Moloney murine leukemia viruses specifically recombine with different endogenous retroviral sequences to generate mink cell focus-forming viruses. *Proc Natl Acad Sci U S A* **82**, 459-463 (1985).
45. Jern, P., Stoye, J.P. & Coffin, J.M. Role of APOBEC3 in genetic diversity among endogenous murine leukemia viruses. *PLoS Genet* **3**, 2014-2022 (2007).
46. Bamunusinghe, D. *et al.* Recombinant Origins of Pathogenic and Nonpathogenic Mouse Gammaretroviruses with Polytopic Host Range. *J Virol* **91** (2017).
47. Chattopadhyay, S.K., Lowy, D.R., Teich, N.M., Levine, A.S. & Rowe, W.P. Evidence that the AKR murine-leukemia-virus genome is complete in DNA of the high-virus AKR mouse and incomplete in the DNA of the "virus-negative" NIH mouse. *Proc Natl Acad Sci U S A* **71**, 167-171 (1974).
48. Taylor, C.S., Nouri, A., Lee, C.G., Kozak, C. & Kabat, D. Cloning and characterization of a cell surface receptor for xenotropic and polytopic murine leukemia viruses. *Proc Natl Acad Sci U S A* **96**, 927-932 (1999).
49. Levy, J.A. Host range of murine xenotropic virus: replication in avian cells. *Nature* **253**, 140-142 (1975).
50. Opstelten, D.J., Wallin, M. & Garoff, H. Moloney murine leukemia virus envelope protein subunits, gp70 and Pr15E, form a stable disulfide-linked complex. *J Virol* **72**, 6537-6545 (1998).

REVIEWERS' COMMENTS

Reviewer #1 (Remarks to the Author):

In the revised version of their manuscript the authors have thoroughly addressed all my concerns and comments and therefore the manuscript has been greatly improved. Thank you very much for your efforts. I also appreciate the numbers of additional- and control experiments that have been carried out for this revision. However, I still disagree with restricting to HERV-K related proteins and particularly excluding the pathogenic variant of HERV-W (pV14). Receptor binding of this protein has not been fully addressed leaving the possibility that other potential receptors such as TLR2, CD14, MFSD2, ASCT1/Slc1a4, ASCT2/Slc1a5 or MCT-1/Slc16a1 are additionally involved. Moreover, activation of this entity is no more restricted to MS and apart from published data on BP, SZ, CIDP and T1D, most likely more pathologies will be described in the near future the corresponding envelope protein is expressed. It would therefore be nice to at least include these thoughts in your discussion, all the more as we are constantly learning about the roles and implications of HERV elements. Moreover, to my understanding our cited reference No. 17 (Schmitt et al., 2013) cannot be used to prove ERVW1/Syncytin1 expression in the CNS, as it is a HERV-K specific paper. You might have had Antony et al., 2004 in mind, but by now it is no more clear whether in their study only Syncytin1 was detected or whether this included HERV-W (pV14). Please consider this information in an amended discussion.

Reviewer #2 (Remarks to the Author):

All the raised issues have been addressed satisfactorily. I do enthusiastically endorse publication of this study.

Reviewer #3 (Remarks to the Author):

All my point were addressed in the revision of the manuscript. The manuscript has improved significantly. Congratulations to the authors.

POINT-BY-POINT RESPONSE TO REVIEWERS

Liu et al. (# NCOMMS-22-24759B):

“Reactivated endogenous retroviruses promote protein aggregate spreading“

We would like to thank the reviewers for compliments on our work and welcome the opportunity to submit a revised version of our work.

Please find enclosed a point-by-point response to the reviewers’ comments.

Reviewer #1 (Remarks to the Author):

In the revised version of their manuscript the authors have thoroughly addressed all my concerns and comments and therefore the manuscript has been greatly improved. Thank you very much for your efforts. I also appreciate the numbers of additional- and control experiments that have been carried out for this revision. However, I still disagree with restricting to HERV-K related proteins and particularly excluding the pathogenic variant of HERV-W (pV14). Receptor binding of this protein has not been fully addressed leaving the possibility that other potential receptors such as TLR2, CD14, MFSD2, ASCT1/Slc1a4, ASCT2/Slc1a5 or MCT-1/Slc16a1 are additionally involved. Moreover, activation of this entity is no more restricted to MS and apart from published data on BP, SZ, CIDP and T1D, most likely more pathologies will be described in the near future the corresponding envelope protein is expressed. It would therefore be nice to at least include these thoughts in your discussion, all the more as we are constantly learning about the roles and implications of HERV elements. Moreover, to my understanding our cited reference No. 17 (Schmitt et al., 2013) cannot be used to prove ERVW1/Syncytin1 expression in the CNS, as it is a HERV-K specific paper. You might have had Antony et al., 2004 in mind, but by now it is no more clear whether in their study only Syncytin1 was detected or whether this included HERV-W (pV14). Please consider this information in an amended discussion.

Response: Thank you very much for acknowledging improvement of our manuscript. As suggested, we now reference the article on the secreted HERV-W Env variant pV14 in the discussion section: “Interestingly, HERV-W Env variants expressed in Multiple Sklerosis patients can also be secreted as soluble hexamers⁶⁶. Their role in other neurodegenerative diseases such as AD remains to be established.” Please understand we had to massively reduce citations (from 111 to 84)- thus we are unfortunately unable to comprehensively cite other studies on this variant.

Reviewer #2 (Remarks to the Author):

All the raised issues have been addressed satisfactorily. I do enthusiastically endorse publication of this study.

Response: Thank you very much.

Reviewer #3 (Remarks to the Author):

All my points were addressed in the revision of the manuscript. The manuscript has improved significantly. Congratulations to the authors.

Response: Thank you very much.